# Impaired yolk sac NAD metabolism disrupts murine embryogenesis with relevance to human birth defects

Kayleigh Bozon[1], Hartmut Cuny[1,2], Delicia Z Sheng[1], Ella MMA Martin[1], Alena Sipka[1], Paul Young[1], David T Humphreys[1,2], Sally L Dunwoodie[1,2]*

[1]Victor Chang Cardiac Research Institute, Sydney, Australia; [2]School of Clinical Medicine, Faculty of Medicine and Health, University of New South Wales, Sydney, Australia

*For correspondence:
s.dunwoodie@victorchang.edu.au

Competing interest: The authors declare that no competing interests exist.

## eLife Assessment

NAD deficiency perturbs embryonic development resulting in multiple congenital malformations, collectively termed Congenital NAD Deficiency Disorder (CNDD). The authors report **fundamental** findings demonstrating that extra-embryonic visceral yolk sac endoderm is critical for NAD de novo synthesis during early organogenesis and perturbations of this pathway may underlie CNDD. The authors combine gene expression with metabolic assays to provide **solid** evidence of an essential role of the extra-embryonic visceral yolk sac in both mouse and human embryos.

**Abstract** Congenital malformations can originate from numerous genetic or non-genetic factors but in most cases the causes are unknown. Genetic disruption of nicotinamide adenine dinucleotide (NAD) de novo synthesis causes multiple malformations, collectively termed Congenital NAD Deficiency Disorder (CNDD), highlighting the necessity of this pathway during embryogenesis. Previous work in mice shows that NAD deficiency perturbs embryonic development specifically when organs are forming. While the pathway is predominantly active in the liver postnatally, the site of activity prior to and during organogenesis is unknown. Here, we used a mouse model of human CNDD and assessed pathway functionality in embryonic livers and extraembryonic tissues via gene expression, enzyme activity and metabolic analyses. We found that the extra-embryonic visceral yolk sac endoderm exclusively synthesises NAD de novo during early organogenesis before the embryonic liver takes over this function. Under CNDD-inducing conditions, visceral yolk sacs had reduced NAD levels and altered NAD-related metabolic profiles, affecting embryo metabolism. Expression of requisite pathway genes is conserved in the equivalent yolk sac cell type in humans. Our findings show that visceral yolk sac-mediated NAD de novo synthesis activity is essential for mouse embryogenesis and its perturbation causes CNDD. As mouse and human yolk sacs are functionally homologous, our data improve the understanding of human congenital malformation causation.

## Introduction

Congenital malformations with substantial medical consequences are a major health burden because of their severity and high prevalence, occurring in approximately 3% of live births (*Moorthie et al., 2018*). Causes of congenital malformation are complex and diverse and can involve genetic as well as environmental factors, complicating the identification of aetiologies. Consequently, the underlying causes of malformation are still unknown in the majority of cases.

Disruption of the nicotinamide adenine dinucleotide (NAD) de novo Synthesis Pathway, by which L-tryptophan is converted to NAD (*Figure 1A*), causes congenital malformations of varying types and severity. It affects organs and structures such as heart, kidney, vertebrae and others, and is termed Congenital NAD Deficiency Disorder (CNDD; *Dunwoodie et al., 2023*). NAD, referring to its oxidised and reduced form (NAD⁺ and NADH), is both a critical redox cofactor present in all living cells and an essential co-enzyme involved in a diverse range of cellular functions. Besides NAD de novo synthesis from L-tryptophan, mammals also generate NAD from vitamin B3, a collective term for the vitamers nicotinic acid (NA), nicotinamide (NAM), nicotinamide mononucleotide (NMN), and nicotinamide riboside (NR), via the Preiss-Handler and NAD Salvage Pathways (*Figure 1A*).

In all CNDD cases identified to date, affected patients had biallelic pathogenic variants in either *KYNU* (kynureninase), *HAAO* (3-hydroxyanthranilic acid 3,4-dioxygenase) or *NADSYN1* (NAD synthetase 1), all essential genes of NAD de novo synthesis (*Dunwoodie et al., 2023*; *Shi et al., 2017*; *Szot et al., 2020*; *Szot et al., 2024*). *Kynu*, *Haao*, or *Nadsyn1* homozygous-null mouse models in combination with diets low in vitamin B3 recapitulated the human phenotypes (*Shi et al., 2017*; *Szot et al., 2024*). When maternal vitamin B3 provision was restricted during pregnancy, it induced CNDD-like defects specifically in homozygous-null mouse embryos, and these embryos had no malformations when the mother was provided sufficient dietary vitamin B3 to bypass their genetic block, as here NAD is synthesised via the Preiss-Handler Pathway or NAD Salvage Pathway (*Figure 1A*; *Shi et al., 2017*). Conversely, embryos of wild-type mothers given diets low in L-tryptophan and depleted in vitamin B3 exhibit NAD deficiency and CNDD-like malformations, with phenotypic severity exacerbated by gene-environment interactions via *Haao* or *Nadsyn1* heterozygosity (*Cuny et al., 2020*; *Szot et al., 2024*).

These CNDD mouse models suggest that NAD de novo synthesis is active in the conceptus and is required for normal organ development unless sufficient vitamin B3 is supplied by the mother. In adult mice, NAD de novo synthesis occurs in the liver and to a very minor extent in the kidney (*Liu et al., 2018*; *Wang et al., 2023*). But mouse embryos are susceptible to CNDD between embryonic day (E) 7.5 and E12.5, before the embryo has developed a functional liver. Therefore, NAD de novo synthesis is likely active in another tissue during this time window of susceptibility.

The major components of the mammalian conceptus are the embryo itself and two extraembryonic organs, the chorioallantoic placenta and the visceral yolk sac (hereon yolk sac). Perturbation of either extraembryonic organ function can induce malformations in the embryo (*Brent and Fawcett, 1998*; *Brent et al., 1971*; *Perez-Garcia et al., 2018*). Therefore, to determine where NAD de novo synthesis activity occurs in the conceptus, we assessed pathway function in these extraembryonic tissues during the CNDD-susceptible organogenesis stages, and in the embryonic liver once formed. We quantified expression of NAD de novo Synthesis Pathway genes, HAAO enzyme activity, and metabolites formed during the synthesis and consumption of NAD. We identified yolk sac endoderm cells as the site of NAD de novo synthesis during organogenesis in mice. A comparison of mouse and human single-cell RNA datasets indicates that these findings in mouse are relevant to humans.

## Results

### NAD de novo synthesis in the embryonic liver is active after organogenesis in mice

NAD is synthesised de novo from L-tryptophan, predominantly in the liver. In addition, all cells salvage NAD from NAM in the circulation, with circulatory NAM levels dependent on NAD de novo synthesis in the liver (*Liu et al., 2018*). We have shown in mice that the conceptus is able to synthesise NAD de novo prior to E12.5 *Shi et al., 2017*; however, it is unclear if or when the embryonic liver is capable of NAD de novo synthesis. In mice, liver primordial cells are present around E9.5 to E10.5, hepatocytes are present from E11.5, and enzymes of the pathway are detected from E13.5 (*Burke et al., 2018*; *Mu et al., 2020*; *Notenboom et al., 1997*; *Yang et al., 2017*).

To determine when NAD de novo Synthesis Pathway activity commences in hepatocytes during embryogenesis, we re-analysed published murine single-cell RNA-seq (scRNA-seq) data for the hepatic lineage (liver primordium, liver bud, and hepatocytes; *Mu et al., 2020*; *Figure 1B*). *Tdo2* and *Ido2*, encoding tryptophan-catabolising enzymes, were negligibly expressed at all stages. Expression of subsequent pathway genes *Afmid*, *Kmo*, *Haao*, *Qprt* and *Nadsyn1*, required for NAD de

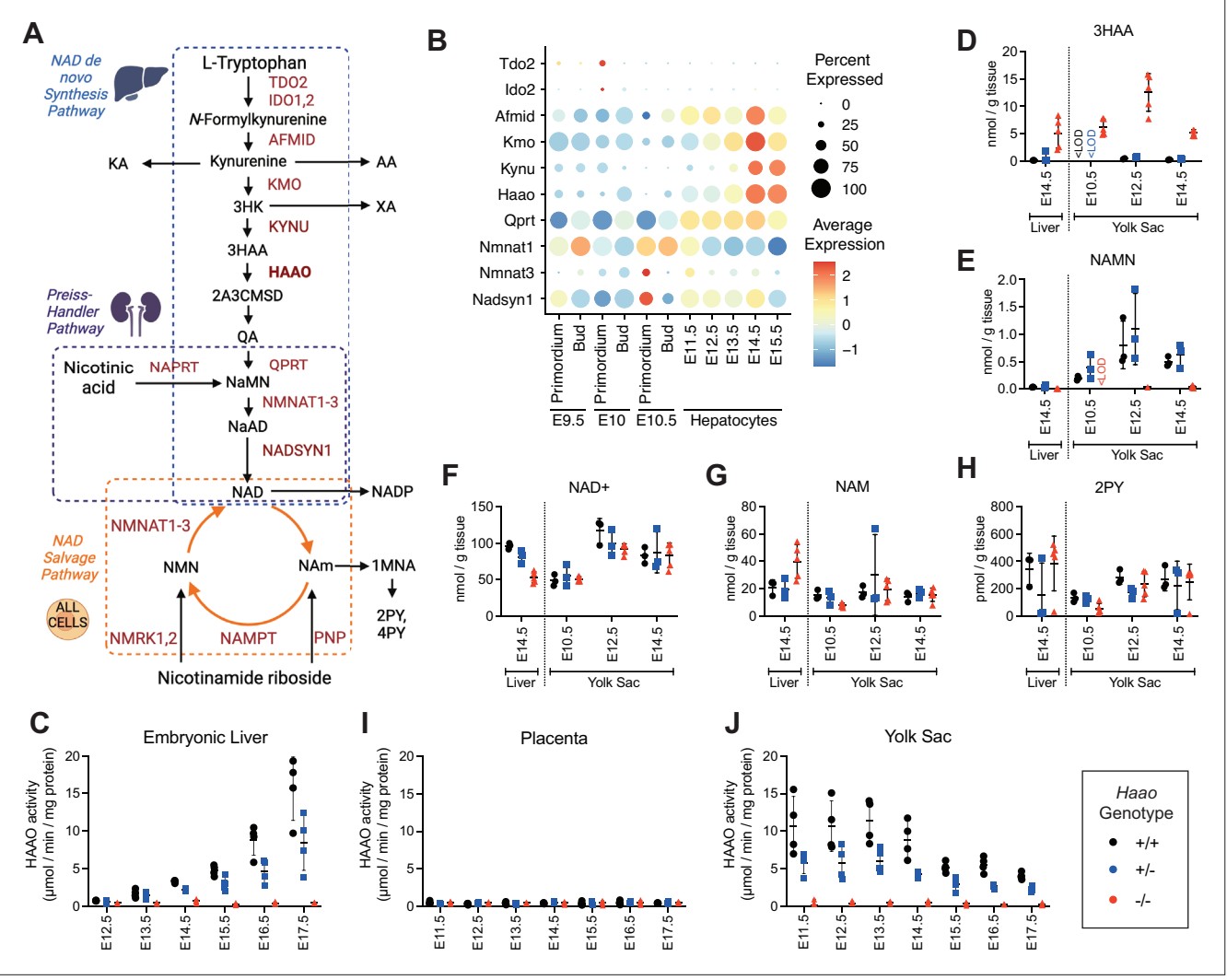

**Figure 1.** NAD de novo synthesis is performed by the yolk sac during organogenesis and later by the embryonic liver. (**A**) Schematic overview of the NAD synthesis pathways. The NAD de novo Synthesis Pathway (blue box) converts the essential amino acid L-tryptophan to NAD and is predominantly performed by the liver (*Liu et al., 2018*). The Preiss-Handler Pathway (purple box) converts nicotinic acid to NAD in the kidney. The Salvage Pathway (orange box), performed in all cells, salvages the product of NAD consuming enzymes, nicotinamide (NAM), back to NAD. When NAD levels are in excess, NAM is converted to 1-methylnicotinamide (1MNA), *N*-methyl-2-pyridone-5-carboxamide (2PY) and *N*-methyl-4-pyridone-5-carboxamide (4PY) and excreted. Enzymes are in red, metabolites in black. (**B**) Re-analysis of published scRNA-seq data (*Mu et al., 2020*) via DotPlot showing average log normalised expression (heatmap colour) and proportion of cells (circle size) of requisite NAD de novo Synthesis Pathway genes in the hepatocyte lineage between embryonic day 9.5 (E9.5) and E15.5. (**C**) HAAO enzyme activity in embryonic liver from E12.5 to E17.5. (**D–H**) Concentration of core NAD de novo Synthesis Pathway metabolites in the embryonic liver (E14.5) and yolk sac (E10.5–14.5). Metabolite concentrations are normalised to wet weight measured at dissection. (**I,J**) HAAO enzyme activity in placenta (**I**) and visceral yolk sac (**J**) samples collected between E11.5 to E17.5. *Haao*$^{+/+}$ (black circle), *Haao*$^{+/-}$ (blue square) and *Haao*$^{-/-}$ (red triangle). Bars indicate the mean ± standard deviation. See *Supplementary file 1* for HAAO activity numerical values and *Supplementary file 2* and *Supplementary file 3* for metabolite concentration values, including those of additional NAD-related metabolites. 3HK = 3-hydroxykynurenine, 3HAA = 3- hydroxyanthranilic acid, NAMN = nicotinic acid mononucleotide, KA = kynurenic acid, AA = anthranilic acid, XA = xanthurenic acid, QA = quinolinic acid, NaMN = nicotinic acid mononucleotide, NaAD = nicotinic acid adenine dinucleotide, NADP = nicotinamide adenine dinucleotide phosphate, NMN = nicotinamide mononucleotide, <LOD = below the limit of detection.

The online version of this article includes the following figure supplement(s) for figure 1:

**Figure supplement 1.** RidgePlots show that the time of onset, expression level, and proportion of hepatocyte-lineage cells expressing NAD de novo synthesis genes varies, including for *Kmo* (**A**), *Kynu* (**B**), *Haao* (**C**) and *Qprt* (**D**) from E9.5 to E15.5.

**Figure supplement 2.** The mouse embryo has negligible HAAO enzyme activity and thus lacks NAD de novo synthesis capability until it has developed a liver.

**Figure supplement 3.** NAD de novo Synthesis Pathway activity is absent from the mouse placenta.

novo synthesis from kynurenine (also referred to as the kynurenine pathway), progressively increased during embryogenesis (*Figure 1B*, *Figure 1—figure supplement 1*), though the time of onset and proportion of cells varied between genes. Nonetheless, all were expressed by E14.5 (*Figure 1B*). HAAO enzyme assays were performed on *Haao*[+/+], *Haao*[+/-] and *Haao*[-/-] embryonic livers, isolated from mothers provided a diet rich in L-tryptophan and vitamin B3 (Breeder diet, see Methods). The livers showed limited HAAO activity at E12.5 and a gradually increase at subsequent developmental stages. Furthermore, HAAO activity correlated with the number of functional *Haao* alleles and was significantly different between genotypes at all stages assessed (*Figure 1C*, *Supplementary file 1*). Whole E11.5 embryos as well as E14.5 embryos with their livers removed had negligible HAAO enzymatic activity (*Figure 1—figure supplement 2*, *Supplementary file 1*), confirming that no other embryonic tissue apart from the liver is capable of NAD de novo synthesis.

To validate that the pathway is functional at E14.5 in the liver, we quantified NAD[+] and 11 NAD-related metabolites in *Haao*[+/+], *Haao*[+/-] and *Haao*[-/-] liver samples using ultra-high-performance liquid chromatography–tandem mass spectrometry (UHPLC-MS/MS; *Cuny et al., 2021*; *Supplementary file 2*). Differential accumulation of metabolites upstream and downstream of the *Haao* genetic block is a defining characteristic of enzyme loss-of-function in both mice and humans (*Shi et al., 2017*; *Szot et al., 2024*). Accordingly, *Haao*[-/-] E14.5 livers had significantly elevated levels of the upstream metabolite 3HAA (18-fold) relative to *Haao*[+/+], whereas the downstream metabolites QA and NAMN were undetectable and significantly decreased (fourfold), respectively (*Figure 1D and E*, *Supplementary file 2*). NAD[+] levels were also significantly affected by the loss of functional *Haao* alleles (*Figure 1F*, *Supplementary file 2*), with the lowest concentration measured in *Haao*[-/-] embryonic livers. By contrast, levels of the NAD Salvage Pathway metabolite NAM were 2-fold higher in *Haao*[-/-], compared to *Haao*[+/+] and *Haao*[+/-] livers (*Figure 1G*, *Supplementary file 2*) and levels of the waste products *N*-methyl-2-pyridone-5-carboxamide (2PY) and *N*-methyl-4-pyridone-3-carboxamide (4PY) were similar between *Haao* genotypes (*Figure 1H*, *Supplementary file 2*).This suggests that NAD Salvage Pathway activity compensated for the loss of NAD de novo synthesis activity in *Haao*[-/-] embryos.

Together, these data confirm that NAD de novo Synthesis Pathway gene expression occurs in the developing liver from E12.5 onwards and the pathway is functional by E14.5, which is in line with expression of genes encoding other metabolic enzymes (*Mu et al., 2020*). However, as CNDD malformations occur in *Haao*[-/-] mouse embryos when NAD is deficient between E7.5 and E12.5 (*Shi et al., 2017*), NAD de novo synthesis must occur elsewhere in the earlier-stage conceptus.

## NAD de novo synthesis occurs during early organogenesis in the mouse yolk sac but not the placenta

In mammals, two likely candidate sites for extrahepatic NAD de novo synthesis are the chorioallantoic placenta and the yolk sac. First, we re-analysed published mouse E9.5–14.5 placenta scRNAseq data (*Marsh and Blelloch, 2020*) and found that NAD de novo Synthesis Pathway gene expression was scattered between the maternal decidual stroma (*Tdo2*, *Ido1*, *Ido2*) or various immune cell populations (*Afmid*, *Kmo*, *Kynu*, *Haao*, *Qprt*, *Nadsyn1*) but not in the conceptus-derived trophoblasts that constitute the bulk of placental tissue (*Figure 1—figure supplement 3*). Additionally, we showed that HAAO enzyme activity was absent in the mouse placenta between E11.5 and E17.5 (*Figure 1I*, *Supplementary file 1*). Combined, these data indicate that the murine placenta does not conduct NAD de novo synthesis.

Next, we quantified HAAO enzyme activity in the yolk sac between E11.5 and E17.5. By contrast to the placenta, *Haao*[+/+] and *Haao*[+/-] yolk sacs exhibited HAAO enzyme activity during early organogenesis (E11.5 to E13.5), after which it progressively declined (*Figure 1J*, *Supplementary file 1*). The decline in HAAO activity in the yolk sac coincided with its rise in the embryonic liver (*Figure 1C*, *Supplementary file 1*). In the absence of available stage-appropriate RNA-seq datasets for the yolk sac, we assessed NAD de novo Synthesis Pathway gene expression by RT-qPCR between E10.5 and E17.5. Transcript levels for early pathway genes (*Tdo2*, *Ido2*, *Afmid*) were similar across all stages, whereas expression of subsequent genes (*Kmo*, *Kynu*, *Haao*, *Qprt*, *Nadsyn1*) was highest at E10.5, then declined 2–10 fold by E17.5 (*Figure 2A and B*).

We quantified NAD[+] and 10 NAD-related metabolites in *Haao*[+/+], *Haao*[+/-] and *Haao*[-/-] yolk sac samples at E10.5, E12.5, and E14.5. Metabolites spanning the entire NAD de novo Synthesis Pathway were quantifiable at all stages (*Supplementary file 3*). The yolk sac metabolite data were compared

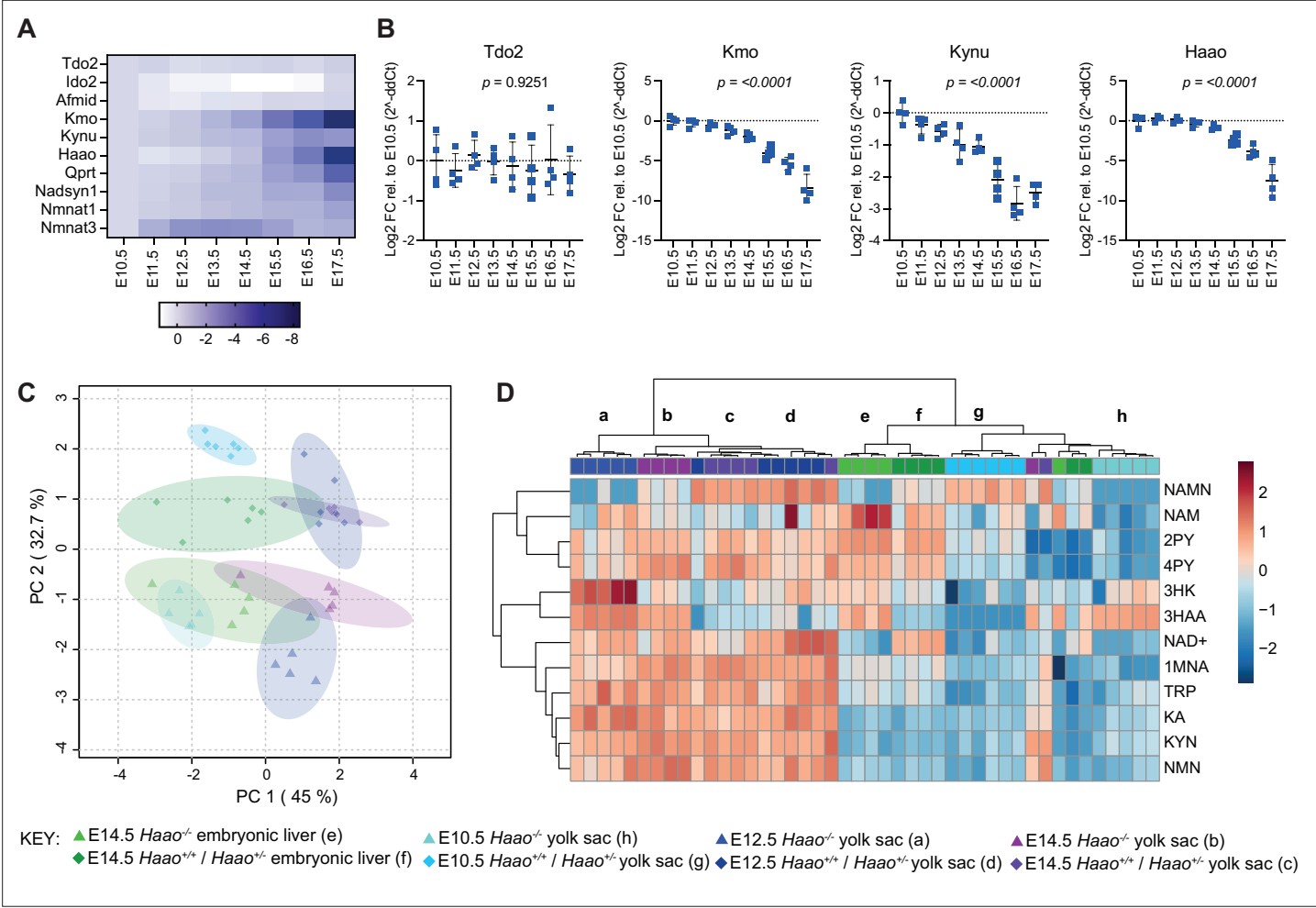

**Figure 2.** The yolk sac performs NAD de novo synthesis from E10.5 onwards. (**A–B**) Heatmap (**A**) and bar graphs (**B**) showing changes in core NAD de novo NAD Synthesis Pathway gene expression in *Haao*[+/-] visceral yolk sacs between E10.5 and E17.5 quantified by RT-qPCR (n=4). Expression is calculated as log2 fold-changes relative to gene expression at E10.5 and normalised to two housekeeping genes, *Ubc* and *Ywhaz*. Bars indicate the mean ± standard deviation. Statistical significance was determined by one-way ANOVA. (**C–D**) NAD-related metabolite concentrations in visceral yolk sacs at embryonic day 10.5 (E10.5), E12.5 and E14.5 were compared to metabolite concentrations in E14.5 embryonic livers. (**C**) Principal component analysis (PCA) of the concentrations of NAD$^+$ and 11 related metabolites in the three tissues at each embryonic stage. Embryonic liver (green) and yolk sac (light blue, indigo, purple); triangles: *Haao*[-/-] tissues, diamonds: *Haao*[+/+] and *Haao*[+/-] tissues. The primary determinants of variance in the NAD metabolome were gestational age (PC 1) and *Haao* genotype (PC 2). (**D**) Heatmap showing wet weight normalised, log10 transformed, and Pareto scaled concentration of each quantified metabolite across all samples, tissues and timepoints. Clusters are labeled with lowercase letters corresponding to tissues and genotypes specified in the key. Consistent with the PCA, the metabolite profile of E10.5 yolk sacs clustered to that of E14.5 embryonic liver. Hierarchical clustering was performed using Euclidean as distance measure and Ward as a clustering method. See ***Supplementary file 2*** and ***Supplementary file 3*** for numerical metabolite concentration values.

to the E14.5 embryonic liver by principal component analysis, which showed gestational age and *Haao*[-/-] genotype as the main determinants of variance in the NAD metabolism (***Figure 2C***). Metabolite levels, normalised to tissue weight, were generally low in E10.5 yolk sacs, similar to levels of E14.5 embryonic liver, and generally higher in E12.5 and E14.5 yolk sacs (***Figure 2D***, ***Supplementary file 3***). *Haao*[-/-] yolk sacs also exhibited the metabolic signature characteristic of HAAO genetic block, with significant accumulation (~20-fold) of upstream 3HAA, and absence of the downstream NAMN relative to *Haao*[+/+] and *Haao*[+/-] yolk sacs at all stages (***Figure 1D and E***, ***Supplementary file 3***). Downstream of NAMN, NAD$^+$ levels were not significantly affected by *Haao* genotype at any stage (***Figure 1F***, ***Supplementary file 3***). However, levels of the NAD Salvage Pathway metabolite NAM and waste metabolites 2PY and 4PY were significantly reduced in E10.5 *Haao*[-/-] yolk sacs relative to

*Haao*<sup>+/+</sup> and *Haao*<sup>+/-</sup> (**Figure 1G and H**, **Supplementary file 3**), suggesting that loss of NAD de novo synthesis resulted in a metabolic adjustment of the NAD Salvage Pathway to maintain NAD levels.

These data, combined with the presence of HAAO activity and expression of all requisite pathway genes, confirmed that the yolk sac performs NAD de novo synthesis from at least E10.5 onwards and therefore is the site of NAD de novo Synthesis Pathway activity in the conceptus during organogenesis prior to the formation of a functional liver.

## NAD deficiency between E7.5 and E10.5 induces CNDD-related congenital anomalies in mice

Having confirmed that NAD de novo synthesis is performed in the yolk sac during early organogenesis, we next sought to understand how its activity is perturbed under conditions that cause CNDD. The data outlined above was generated using a diet rich in vitamin B3 which allows pregnant mice with a genetic block in the NAD de novo Synthesis Pathway to generate NAD via the Preiss-Handler and NAD Salvage Pathway and provide sufficient NAD precursors to their embryos for normal development. Due to higher metabolic rate and food intake per body weight in mice compared to humans (**Bachmanov et al., 2002**; **Holliday et al., 1967**), to induce CNDD in mice, pregnant females need to be given L-tryptophan and/or vitamin B3 restricted diets, thereby limiting their ability to provide NAD precursors to the embryos (**Cuny et al., 2020**; **Shi et al., 2017**; **Szot et al., 2024**). Given that

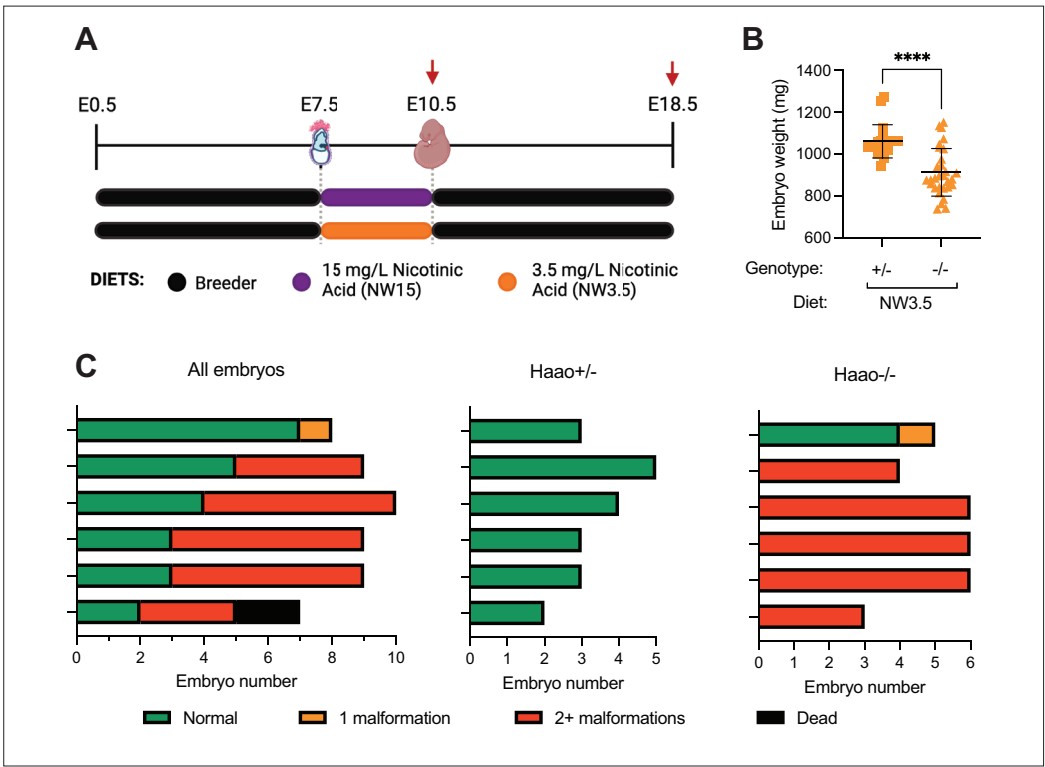

**Figure 3.** Congenital NAD Deficiency Disorder (CNDD) phenotypes and NAD deficiency can be induced in *Haao*<sup>-/-</sup> embryos by restricting maternal NAD precursor supply for three days. (**A**) Schematic overview of the dietary treatment to induce NAD deficiency and embryo malformations. Pregnant mice were provided Breeder diet, rich in L-tryptophan and vitamin B3, throughout gestation except for a three-day window between embryonic day 7.5 (E7.5) and E10.5 during which feed lacking any NAD-precursor and drinking water supplemented with 1 g/L L-tryptophan and either 15 mg/L (NW15) or 3.5 mg/L (NW3.5) nicotinic acid was given. Red arrows indicate tissue collection timepoints. (**B**) Weights of *Haao*<sup>+/-</sup> (square) or *Haao*<sup>-/-</sup> (triangle) embryos at E18.5 collected from mothers on NW3.5 diet. Weights represent wet weight measured at dissection. Statistical significance was determined by Student's t-test with **** p<0.0001. Bars indicate the mean ± standard deviation. (**C**) Summary of the phenotypic outcomes of E18.5 embryos generated from the NW3.5 diet condition. Each bar represents a litter, with the graph on the left summarising data of entire litters and the other two separating the *Haao*<sup>+/-</sup> and *Haao*<sup>-/-</sup> embryos of each litter. Colours indicate the number of different congenital malformations observed in each embryo. Dead embryos (black) represented early resorptions and could not be genotyped.

NAD de novo synthesis is established in the yolk sac by E10.5, we placed *Haao*-/- mothers on a NAD precursor-restricted diet specifically between E7.5 and E10.5. During this time window, L-tryptophan and vitamin B3 were removed from the chow and 1000 mg/L L-tryptophan and 3.5 mg/L nicotinic acid were supplemented in the drinking water (hereon referred to as NW3.5) (*Figure 3A*). Unlike wild-type and *Haao*+/- mice, *Haao*-/- mice can only use the vitamin B3 but not L-tryptophan in the diet to synthesise NAD. Here, *Haao*-/- females were mated with *Haao*+/- males and generated embryos with *Haao*+/- and *Haao*-/- genotypes, of which only the former can use any L-tryptophan provided by the maternal circulation.

Embryos from mothers on the NW3.5 diet were assessed for the presence of congenital anomalies at E18.5. *Haao*-/- embryos were significantly smaller than their *Haao*+/- littermates (*Figure 3B*). Furthermore, 86.7% of *Haao*-/- embryos were malformed, with 83.3% (25/30) of them presenting with multiple defects (*Figure 3C*, *Supplementary file 4*). We identified malformations in organs and tissues previously associated with CNDD (*Cuny et al., 2020*; *Shi et al., 2017*). Kidney defects (in 83.3% of *Haao*-/- embryos) included hypoplasia or agenesis, as well a cyst observed in one kidney. In addition, shortened tails (73.3%), various structural heart defects (27.7%), digit and limb anomalies (40%), and vertebrae and rib defects (63.3%) including rib underdevelopment, sternum defects, vertebral fusions, butterfly vertebrae, and hemivertebrae were found (*Supplementary file 4*). By contrast, the ability to perform NAD de novo synthesis prevented NAD deficiency in *Haao*+/- littermate embryos; the only defect was a solitary case of heart ventricular septum defect (*Figure 3C*, *Supplementary file 4*).

These observations align with previous CNDD studies utilising *Haao*-/- mothers on a vitamin B3-restricted diet (*Cuny et al., 2020*; *Shi et al., 2017*). In addition, they show that restricting dietary precursor supply for three days only, between E7.5 and E10.5, is sufficient to cause CNDD malformation. This narrow window of susceptibility highlights the necessity of NAD de novo Synthesis Pathway activity in the yolk sac for normal development.

## The yolk sac has reduced NAD availability under conditions that induce congenital malformations

Having established the importance of E10.5 as a key stage of NAD de novo synthesis, we next characterised how maternal dietary NAD precursor-restriction between E7.5 and E10.5 affected levels of NAD and related metabolites in the yolk sac at E10.5. We previously showed that supplementing 15 mg/L nicotinic acid in the drinking water is sufficient to support normal development of *Haao*-/- embryos (*Shi et al., 2017*). Therefore, yolk sac samples from the maternal NW15 dietary condition (*Figure 3A*) were collected as appropriate controls for comparison to NW3.5 samples.

Due to their small size, 2–3 yolk sacs were pooled per sample for NAD metabolite quantification by UHPLC-MS/MS. The diet itself had profound effects on the yolk sac metabolites (*Supplementary file 5* and *Supplementary file 6*). Although L-tryptophan concentration in the drinking water was equivalent in both diets, its levels were lower in NW3.5 yolk sacs than NW15 (*Supplementary file 5* and *Supplementary file 6*, *Figure 4A*). The abundance of NAD de novo Synthesis Pathway metabolites both upstream (3HAA) and downstream (QA, NAMN, NAAD) of the *Haao* genetic block was significantly altered in *Haao*-/- and *Haao*+/- yolk sacs, respectively, relative to their NW15 counterparts (*Figure 4B-E*, *Supplementary file 5* and *Supplementary file 6*). Despite this, NAD+ levels were comparable between *Haao*+/- yolk sacs on both diets (*Figure 4F*, *Supplementary file 5* and *Supplementary file 6*). By contrast, the inability of *Haao*-/- yolk sacs to perform NAD de novo synthesis resulted in a significant decline in NAD+ (54–71%) and NAM (69–77%) levels relative to *Haao*+/- on the equivalent diet (*Figure 4F–G*, *Supplementary file 5* and *Supplementary file 6*). Yet, yolk sac NAD+ levels were lowest when the genetic disruption and dietary restriction were combined (NW3.5 *Haao*-/-), confirming the importance of maternal provision in maintaining NAD+ levels when the yolk sac cannot generate NAD de novo.

## The NAD metabolome of the yolk sac and embryo correlate during early organogenesis

The yolk sac has a nutritional role in mammals as it uptakes and processes maternally derived macro- and micronutrients and transfers these to the embryo via their interconnected and expanding vasculature (*Burton et al., 2016*; *Ornoy and Miller, 2023*). In mice, histotrophic nutrition via the yolk sac is solely responsible for supporting the nutritional requirements of the developing embryo until the

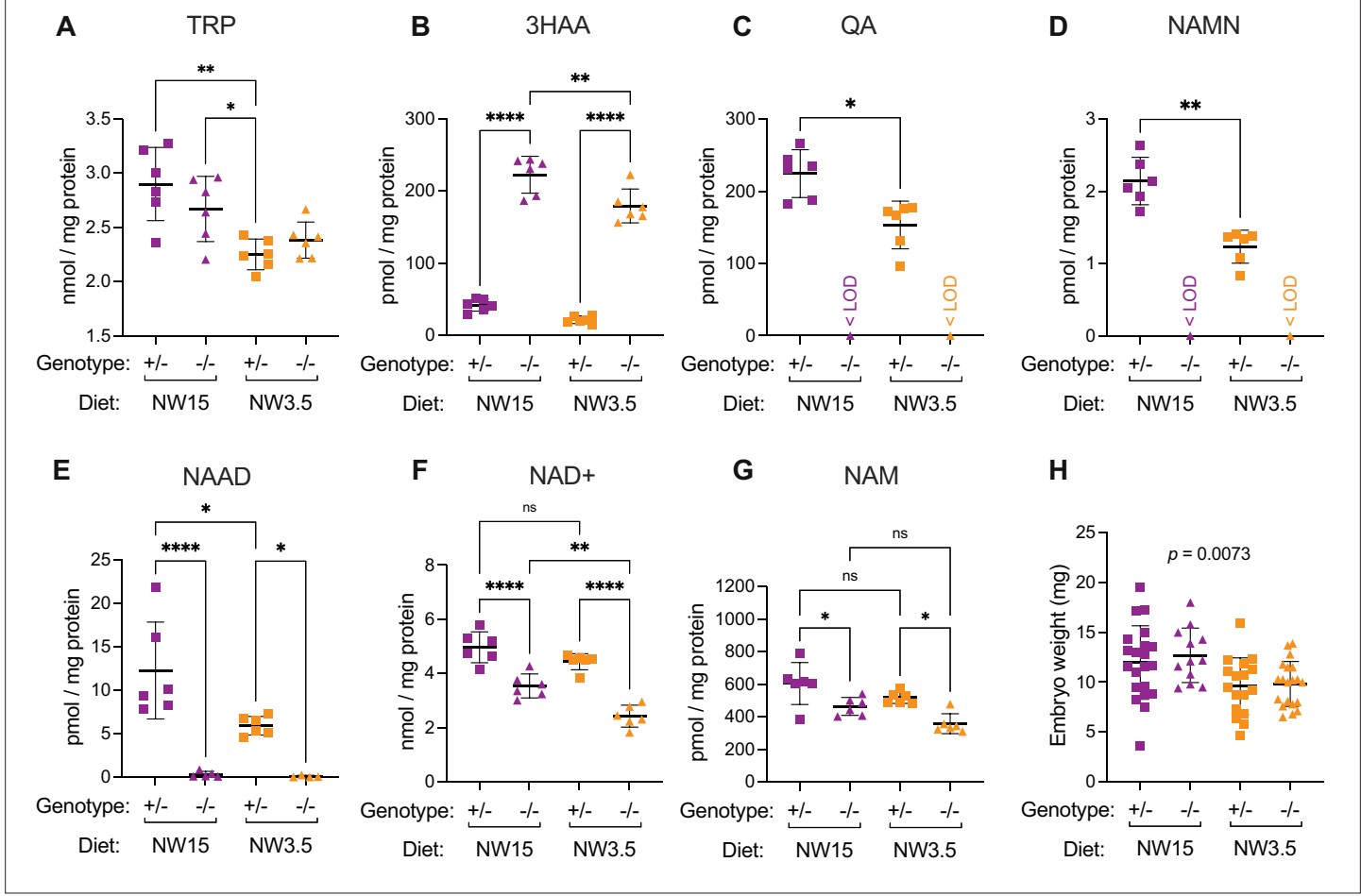

**Figure 4.** The yolk sac NAD metabolome is affected by maternal dietary limitation of NAD precursors that causes embryo malformations. (**A–G**) Concentration of NAD-related metabolites in E10.5 yolk sacs normalised to total protein content. (**A**) L-tryptophan (TRP), (**B**) 3-hydroxyanthranilic acid (3HAA), (**C**) quinolinic acid (QA), (**D**) nicotinic acid mononucleotide (NAMN), (**E**) nicotinic acid adenine dinucleotide (NAAD), (**F**) NAD+, (**G**) nicotinamide (NAM). (**H**) Wet weight of E10.5 embryos from the NW15 (purple) and NW3.5 (orange) diet conditions, measured at dissection. Bars indicate the mean ± standard deviation. The *Haao* genotype and diet are indicated below each graph. Statistical significance in (**A–G**) was calculated by one-way ANOVA with Tukey's multiple comparisons test with ns = not significant, * p<0.05, ** p<0.01, and **** p<0.0001. Statistical significance in (**H**) was calculated by one-way ANOVA with the *p* value indicated above graph. <LOD = below the limit of detection. For numerical values, see **Supplementary file 5**.

gradual transition to haemotrophic nutrition across the chorioallantoic placenta between E10 and E12 (*Elmore et al., 2022*). Therefore, we explored the effects of impaired NAD de novo synthesis of the yolk sac on NAD metabolite levels of the embryo.

We analysed the embryos that corresponded to the metabolically analysed yolk sacs above. Whilst we had to pool 2–3 yolk sacs at E10.5 for analysis, embryos were sufficiently large to be analysed individually with the embryo corresponding to one of the pooled yolk sacs. Contrary to observations at E18.5 (*Figure 3B*), *Haao* genotype had no effect on embryo weight within the same dietary condition, but embryos were significantly smaller on NW3.5 than NW15 (*Figure 4H*). Consistent with the yolk sac samples, the diet had a marked effect on embryonic NAD status, with levels of NAD$^+$ and the waste metabolites 2PY and 4PY significantly reduced in NW3.5 embryos relative to the same *Haao* genotype on the NW15 diet (*Figure 5E, G and H*, *Supplementary file 5*). However, in agreement with phenotypic outcome at E18.5, NAD$^+$ levels of NW3.5 *Haao*$^{-/-}$ embryos were significantly lower than those of *Haao*$^{+/-}$ embryos on the same diet (*Figure 5E*, *Supplementary file 5*).

NAD de novo Synthesis Pathway metabolites were quantifiable in embryos (*Figure 5A–D*, *Supplementary file 5* and *Supplementary file 6*), with differential abundance of metabolite upstream and downstream of the *Haao* genetic block mirroring those observed in the yolk sac. Neither the placenta nor the embryonic liver synthesise NAD de novo at E10.5, suggesting that these metabolites either

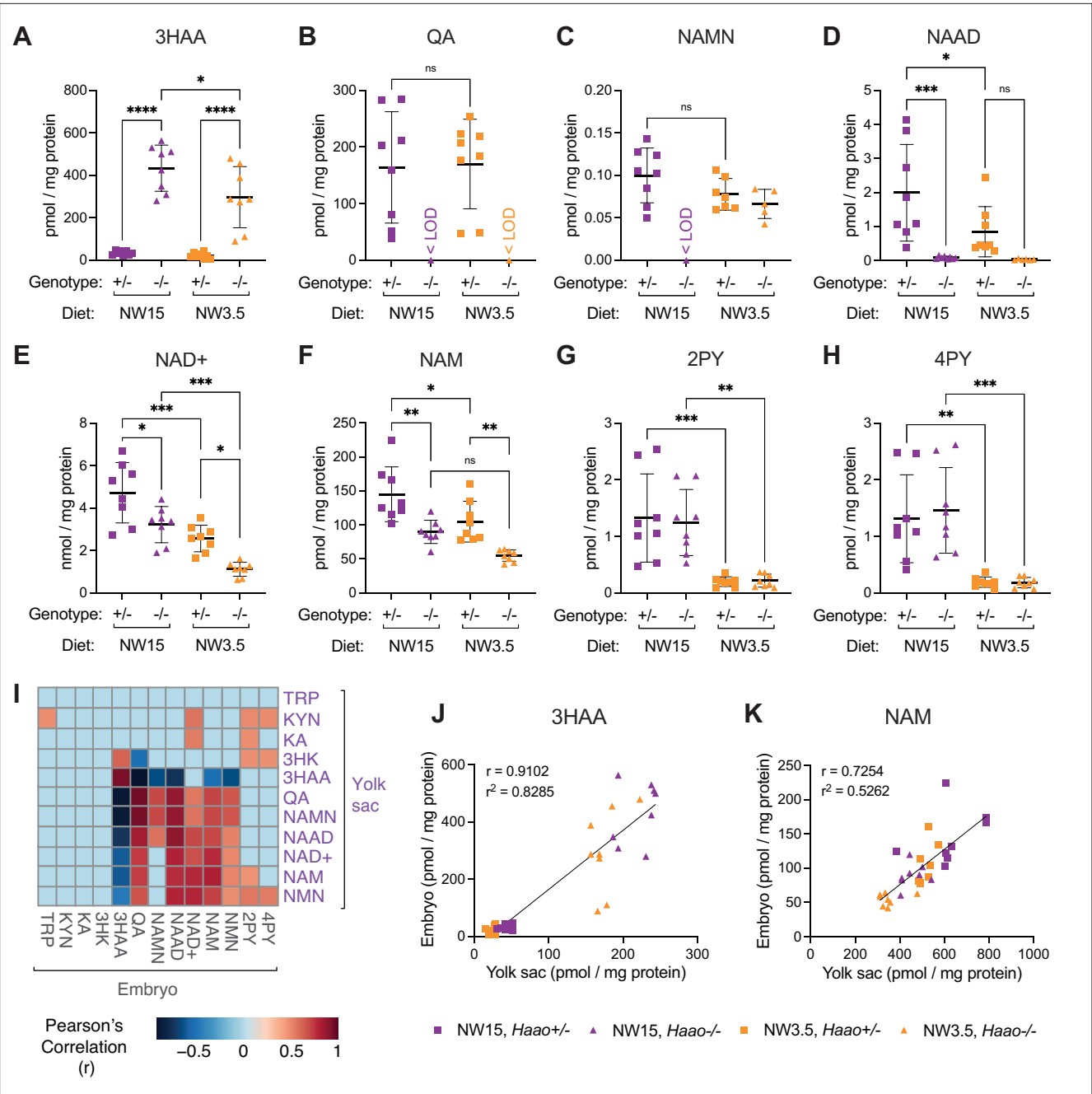

**Figure 5.** Perturbation of yolk sac NAD metabolome at E10.5 correlates with metabolic alterations in the embryo. (A–H) Concentration of NAD-related metabolites in E10.5 embryos normalised to total protein content. (A) 3-hydroxyanthranilic acid (3HAA), (B) quinolinic acid (QA), (C) nicotinic acid mononucleotide (NAMN), (D) nicotinic acid adenine dinucleotide (NAAD), (E) NAD+, (F) nicotinamide (NAM), (G) N-methyl-2-pyridone-5-carboxamide (2PY), (H) N-methyl-4-pyridone-5-carboxamide (4PY). Bars indicate the mean ± standard deviation. Statistical significance in (A–H) was calculated by one-way ANOVA with Tukey's multiple comparisons test with ns = not significant, * p<0.05, ** p<0.01, *** p<0.001, and **** p<0.0001. <LOD = below the limit of detection. For numerical values, see *Supplementary file 6*. (I) Heatmap showing significant Pearson's correlation (p<0.01; –0.45<r >0.45) between normalised metabolite levels in E10.5 yolk sacs and embryos. Correlations that did not reach significance are denoted as light blue coloured squares. (J, K) Correlation of 3HAA (J) and NAM (K) concentration in corresponding E10.5 yolk sac and embryo samples. The Pearson correlation coefficient (r) and coefficient of determination (r²) are indicated in each graph. Each datapoint represents an E10.5 embryo and corresponding pooled yolk sac sample.

The online version of this article includes the following figure supplement(s) for figure 5:

**Figure supplement 1.** NAD de novo Synthesis Pathway gene expression is negligible in the E10.5 embryo.

originate from the yolk sac, which is consistent with the connection of vitelline and embryonic vasculature, or that another embryonic cell type is synthesising NAD de novo at E10.5.

To test the latter option, we examined pathway gene expression in the embryo at E10.5 using the 'mouse organogenesis cell atlas' (MOCA) (*Cao et al., 2019*). Except for some limited *Afmid* and *Kynu* expression predominantly in immune cell populations, the requisite genes were not expressed, including in hepatocytes (*Figure 5—figure supplement 1*). Therefore, intermediary metabolites were seemingly exchanged between the yolk sac and the embryo. To explore this, we calculated the Pearson correlation for all measured NAD-related metabolites between corresponding yolk sacs and embryos. Protein-normalised concentrations of those metabolites differentially abundant by *Haao* genotype (3HAA, NAMN, QA, and NAAD) were significantly positively correlated between the embryo and yolk sac (*Figure 5I–J*). Overall, this suggests extensive metabolite exchange between conceptus tissues occurred.

Levels of NAD Salvage Pathway metabolites NAD⁺, NAM, and NMN were also significantly positively correlated both within and between the yolk sac and embryo tissues (*Figure 5I and K*). NAM is the main NAD precursor exported into circulation after NAD de novo synthesis for use in other tissues (*Liu et al., 2018*). Given the limited availability of other usable alternative NAD precursors in mouse circulation (*Cuny et al., 2021*; *Liu et al., 2018*), NAD levels of *Haao⁻/⁻* conceptuses likely depend on maternal plasma NAM (*Cuny et al., 2023*). Although their overall concentration markedly differed (*Supplementary file 5* and *Supplementary file 6*), embryo and yolk sac NAM correlated strongly, irrespective of *Haao* genotype (*Supplementary file 5*, *Figure 5K*). Therefore, embryonic NAD status depends on NAD precursor availability in the yolk sac, which is itself determined by both maternal provision and NAD de novo synthesis activity.

## The yolk sac endoderm expresses requisite NAD de novo synthesis pathway genes at the onset of organogenesis

Having established the yolk sac as the primary site of NAD de novo synthesis during organogenesis and its importance in maintaining embryonic NAD levels, we next determined which cell populations are responsible for its activity by re-analysing published scRNA-seq data generated from pooled E9.5 and E10.5 yolk sacs (*Figure 6A*; *Zhao and Choi, 2019*).

The yolk sac is derived from two extraembryonic lineages, the endoderm and mesoderm, which subsequently differentiate into various specialised cell types including yolk sac endoderm, vascular endothelial cells, smooth muscle cells, erythroid progenitors/cells, myeloid progenitors/cells, and macrophages (*Zhao and Choi, 2019*). *Tdo2* was only expressed in vascular smooth muscle cells, whereas all other NAD de novo Synthesis Pathway genes (*Afmid, Kmo, Kynu, Haao, Qprt,* and *Nadsyn1*) were exclusively expressed in the yolk sac endoderm (*Figure 6B*). Of the *Nmnat1-3* genes, essential for all NAD biosynthesis pathways, only *Nmnat1* was expressed in the endoderm.

To establish when gene expression commenced in the endodermal lineage, we used the 'extraembryonic (ExE) endoderm' population from the mouse gastrulation atlas (*Pijuan-Sala et al., 2019*). The tryptophan-catabolising genes *Tdo2, Ido1, Ido2* were not expressed in this population at any stage. Expression of all subsequent requisite NAD de novo Synthesis Pathway genes was identifiable in a subset of cells by E8 and increased thereafter (*Figure 6C*). While histological methods such as in situ hybridisation would be required to confirm the exact cell types expressing these genes, the available expression data indicates that the genes encoding those enzymes required to convert L-kynurenine to NAD (kynurenine pathway) are exclusively expressed in the yolk sac endoderm lineage from the onset of organogenesis (E8.0–8.5). Given the lack of *Tdo2* and *Ido2* expression across all timepoints assessed (*Figure 6B and C*), it appears the conversion of tryptophan to kynurenine is either occurring in the yolk sac vascular smooth muscle, or in the placenta which is well known to perform this function in mouse (*Spinelli et al., 2019*) and humans (*Broekhuizen et al., 2021*; *Sedlmayr et al., 2014*).

## NAD de novo synthesis pathway gene expression dynamics in humans parallel those in mice

Despite the structural differences between the mouse yolk sac and its human equivalent, the definitive secondary yolk sac (*Ross and Boroviak, 2020*), there is increasing evidence that many metabolic pathways are conserved in the endodermal cell population (*Cindrova-Davies et al., 2017*; *Goh et al., 2023*). Despite very different timescales of embryogenesis between the two species, the human yolk

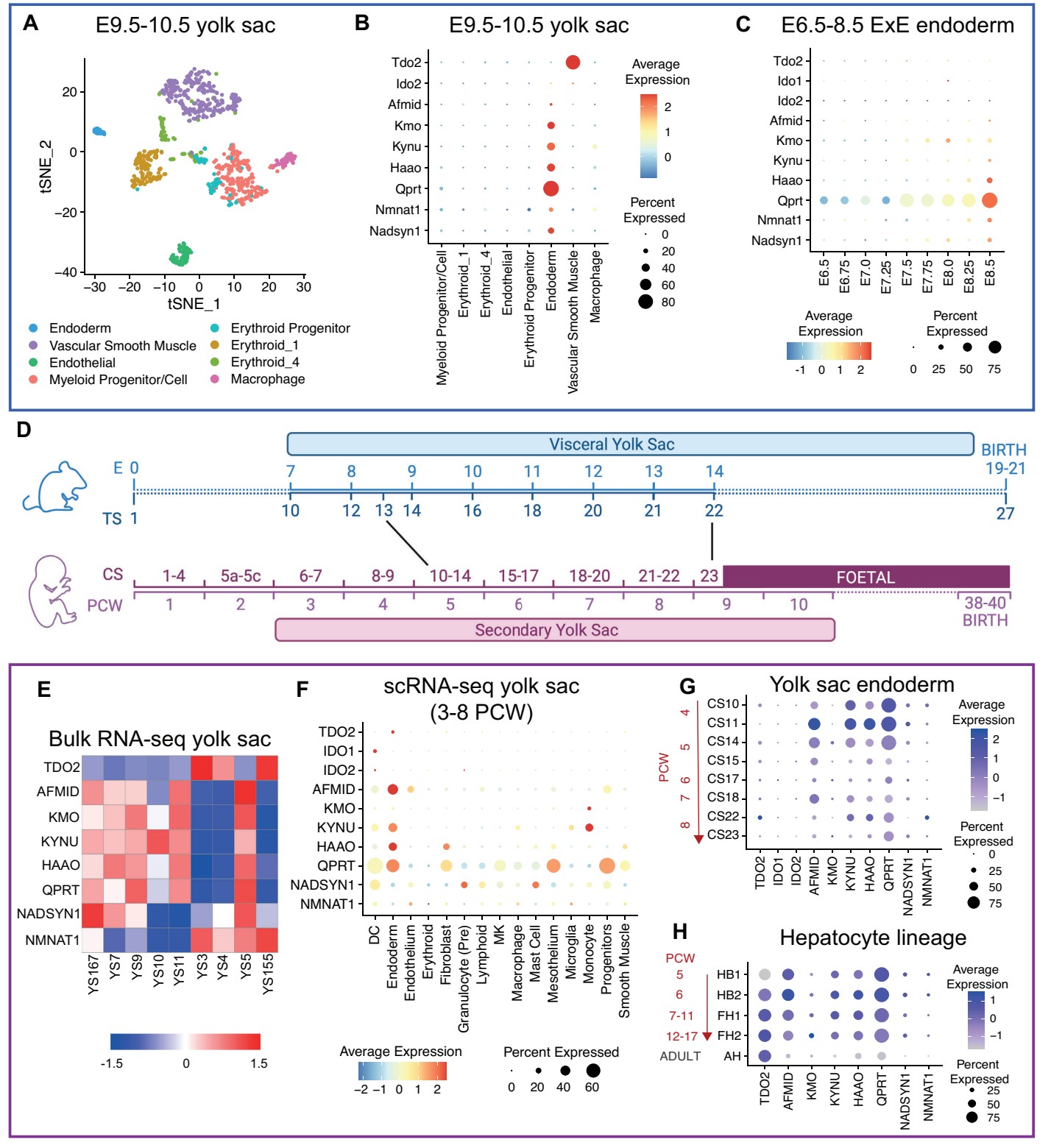

**Figure 6.** The extraembryonic endoderm and subsequent yolk sac endoderm express NAD de novo Synthesis Pathway genes before expression commences in the embryonic liver, in both mouse and human. (**A**) t-SNE projection of re-analysed single-cell RNA-seq data from pooled E9.5 and E10.5 visceral yolk sacs previously generated by *Zhao and Choi, 2019*. (**B**) DotPlot showing average expression (heatmap colour) and proportion of cells (circle size) of essential NAD de novo Synthesis Pathway genes in E9.5-E10.5 yolk sac populations. (**C**) DotPlot for essential NAD de novo Synthesis Pathway genes in the extraembryonic endoderm (ExE) population of the mouse gastrulation atlas (*Pijuan-Sala et al., 2019*) from E6.5 to

*Figure 6 continued on next page*

*Figure 6 continued*

E8.5. (**D**) Schematic timeline comparing organogenesis length and yolk sac duration during mouse and human development. Stages and equivalences are approximations based on figures from https://hdbratlas.org/comparison-HvM.html and published staging criteria (*Kaufman, 1992*; *O'Rahilly and Müller, 2010*) (**E**) Heatmap showing z-normalised expression of NAD de novo Synthesis Pathway genes in re-analysed bulk RNA-seq samples for individual human yolk sacs, published in *Cindrova-Davies et al., 2017*. (**F**) DotPlot for requisite NAD de novo Synthesis Pathway gene expression in human yolk sac populations between 3 and 8 post-conceptional weeks (PCW). scRNA-seq data acquired from *Goh et al., 2023* and re-analysed. (**G**) DotPlot of pathway gene expression in human yolk sac endoderm population between 3 and 8 PCW. (**H**) DotPlot of human hepatocyte lineage previously published in *Wesley et al., 2022*. CS = Carnegie Stage, HB = hepatoblast, FH = foetal hepatocytes, AH = adult hepatocytes.

sac is present throughout organogenesis, the period when embryos are most susceptible to malformation (*Alwan and Chambers, 2015*), after which it degenerates and is lost (*Figure 6D*). Therefore, we investigated whether NAD de novo synthesis is among the conserved metabolic pathways between the two species. We examined expression of the pathway requisite genes by re-analysing bulk and scRNA-seq for the human yolk sac (*Cindrova-Davies et al., 2017*; *Goh et al., 2023*) and hepatocyte lineage (*Wesley et al., 2022*).

All NAD de novo Synthesis Pathway genes were expressed within at least one human yolk sac bulk RNA-seq sample (*Cindrova-Davies et al., 2017*; *Figure 6E*), but there were significant differences in transcript levels between them. High *TDO2* expressing yolk sacs concurrently expressed *NMNAT1*, and to a lesser extent *NADSYN1* (3/9), but showed limited expression of the other requisite genes. Conversely, low *TDO2* expression yolk sacs more robustly expressed *AFMID*, *KMO*, *KYNU*, *HAAO*, *QPRT*, and *NADSYN1* (*Figure 6E*).

Re-analysis of published human scRNA-seq yolk sac data generated across organogenesis (*Goh et al., 2023*) showed that unlike in mice, requisite genes were expressed in multiple populations (*Figure 6F*). *KMO* expression was limited in the endoderm, with more robust expression observed in monocytes. Similarly, *NADSYN1* transcripts were more abundant in various immune cell populations. Yet, expression of *TDO2*, *AFMID*, *KYNU*, *HAAO* and *QPRT* was highest in the endodermal population (*Figure 6F*). As in mice, the proportion of endoderm cells and overall expression of *AFMID*, *KYNU*, *HAAO*, and *QPRT* declined during organogenesis, with expression highest at Carnegie Stage (CS) 11 (4 post-conceptional weeks (PCW)) (*Figure 6G*). *TDO2* expression, not observed in the mouse yolk sac endoderm, increased as organogenesis progressed.

By contrast, in the hepatocyte lineage (*Wesley et al., 2022*), every NAD de novo Synthesis Pathway gene was expressed by 5 PCW (hepatoblast (HB) stages 1 and 2; *Figure 6H*) and remained constant throughout foetal hepatocyte stages (FH) 1 (7–11 PCW) and 2 (12–17 PCW). Indeed, expression during embryonic stages was higher than in adult hepatocytes.

Therefore, requisite NAD de novo synthesis gene expression dynamics in humans largely recapitulate findings in mice with the pathway primarily active in the yolk sac endodermal cells during early organogenesis. As expression declines in this population, it increases in the hepatocyte lineage. The only major difference between the species is whether NAD de novo Synthesis Pathway activity in the embryonic liver is established during or after organogenesis, respectively.

## Discussion

NAD deficiency during human and mouse development causes CNDD characterised by multiple congenital malformations and foetal loss (*Cuny et al., 2023*; *Cuny et al., 2020*; *Dunwoodie et al., 2023*; *Mark and Dunwoodie, 2023*; *Shi et al., 2017*; *Szot et al., 2020*; *Szot et al., 2024*; *Szot et al., 2021*). Human CNDD cases are characterised by biallelic pathogenic variants in either *KYNU*, *HAAO* or *NADSYN1*, all genes of the NAD de novo Synthesis Pathway. Mouse models of CNDD demonstrate that NAD de novo synthesis occurs in the conceptus and that this activity is required for normal embryonic development unless the mother provides sufficient vitamin B3 which bypasses NAD de novo synthesis (*Shi et al., 2017*). Here, we report that NAD de novo synthesis occurs at two sites in the conceptus; first in the transient extra-embryonic yolk sac during organogenesis, and later in the embryonic liver once formed. It is noteworthy that the placenta lacks this activity. We present evidence in human gene expression data that the NAD de novo Synthesis Pathway also occurs initially in the secondary yolk sac and later in the embryonic liver. Until now, the site of NAD de novo synthesis in the

mammalian conceptus was unknown. The necessity of this metabolic activity in the yolk sac highlights the importance of this transient tissue in mammalian embryogenesis.

## The murine yolk sac is capable of synthesising NAD de novo

The yolk sac has various important functions during embryogenesis, including the uptake and processing of macro- and micronutrients before and after the formation of a functional chorioallanotic placenta. It also facilitates the transport of maternal lipids, cholesterol and other metabolites to the embryo until the embryonic liver can take over this function, and it is involved in haematopoiesis and gene regulation (*Cindrova-Davies et al., 2017*). Furthermore, the yolk sac metabolises teratogenic substances (*Terlouw and Bechter, 1992*). Disruption of yolk sac function causes embryonic malformation phenotypes that overlap with CNDD (*Brent and Fawcett, 1998*; *Brent et al., 1971*; *Ornoy and Miller, 2023*; *Perez-Garcia et al., 2018*), underscoring the yolk sac's crucial role for normal embryo development.

Our finding of NAD de novo synthesis activity as another metabolic process of the yolk sac is notable because it represents a multi-step pathway involving several sequential enzyme-catalysed processes. Our RNA-seq data analysis confirms that all requisite genes required for NAD de novo synthesis from kynurenine are expressed in mouse yolk sac endoderm cells. *Tdo2*, required for the conversion of tryptophan to kynurenine, is expressed in the adjacent vascular smooth muscle cells. Our HAAO enzymatic assays and metabolic analyses confirm the pathway is indeed functional. Therefore, this is the first demonstration of NAD de novo synthesis occurring in a tissue outside of the liver and kidney.

## NAD deficiency in early organogenesis is sufficient to cause CNDD

We developed a refined CNDD model by which we modulated the maternal dietary NAD precursor supply during a 3-day window between E7.5 to E10.5 to be either low (NW3.5) or sufficient (NW15). Combined with maternal $Haao^{-/-}$ and paternal $Haao^{+/-}$ genotype, the NW3.5 diet reproducibly generated $Haao^{-/-}$ embryos with malformations and unaffected $Haao^{+/-}$ littermates. The $Haao^{-/-}$ embryos exhibited a spectrum of CNDD defects but no cleft palate seen in our previous study with a precursor restriction for five days between E7.5 and E12.5 (*Shi et al., 2017*). The absence of palate defects is possibly because our current 3-day dietary precursor-restriction is too short to perturb the later occurring developmental processes of palate formation. This observation supports our hypothesis that the timing of NAD deficiency during organogenesis determines which organs/tissues are affected (*Cuny et al., 2023*), but more research is needed to fully characterise the onset and duration of embryonic NAD deficiency in dietary NAD precursor restriction mouse models.

Furthermore, diminished NAD de novo synthesis in the yolk sac is responsible for the defects caused by our 3-day precursor restriction, because the embryonic liver has not yet formed. These data indicate that the timing of NAD deficiency determines whether tissues known to be susceptible to CNDD are affected or not, explaining why phenotypes are highly variable within the CNDD spectrum. But it is currently unknown whether a common underlying molecular mechanism exists that unifies these tissues with respect to their disrupted development in CNDD and whether organs/tissues that are seemingly unaffected by NAD deficiency are resistant to this mechanism. Further research into the molecular and cellular effects of altered NAD availability during organogenesis is required to better understand CNDD causation.

## Maternal vitamin B3 early in pregnancy can overcome a genetic disruption of yolk sac NAD de novo synthesis

The most abundant NAD precursors in circulation are L-tryptophan, L-kynurenine, and NAM (*Cuny et al., 2020*; *Liu et al., 2018*). Flux experiments in mice confirmed that the liver converts L-tryptophan into NAM and excretes it into circulation to maintain NAD levels using the NAD Salvage Pathway (*Liu et al., 2018*). The CNDD models employed here use $Haao^{-/-}$ mothers, who along with their $Haao^{-/-}$ embryos, cannot use the L-tryptophan provided. These $Haao^{-/-}$ mothers and embryos are instead dependent on the vitamin B3 provided in the diet to synthesise NAD. The amount provided in the NW15 diet is sufficient to maintain conceptal NAD levels and normal embryonic development. By contrast, on the more vitamin B3 restricted NW3.5 diet, the ability to perform NAD de novo synthesis

using the L-tryptophan is ultimately what distinguishes the unaffected *Haao*^+/- embryos from their affected *Haao*^-/- littermates.

Indeed, *Haao*^-/- yolk sacs had lowered NAD levels under both diets, but the lowest NAD levels were found in NW3.5 *Haao*^-/- yolk sacs with the combination of genetic block of NAD de novo synthesis and dietary NAD precursor restriction. Similarly, embryonic NAD levels significantly declined under genetic block (*Haao*^-/-, NW15) or vitamin B3 restriction (*Haao*^+/-, NW3.5) alone relative to *Haao*^+/- on the NW15 diet but these were still sufficient to maintain normal development. By contrast, those embryos that would go on to develop defects, *Haao*^-/- embryos on NW3.5, had the lowest NAD levels at E10.5, as observed previously (*Cuny et al., 2020*; *Shi et al., 2017*). This underscores that NAD deficiency in the embryo is linked to the inability of the yolk sac to perform NAD de novo synthesis from maternally provided precursors.

Though unaffected by the *Haao* genetic block, the Preiss-Handler Pathway (*Figure 1A*) does not appear to play a role in yolk sac NAD synthesis given that the relevant metabolites NAMN and NAAD were undetectable or negligible in *Haao*^-/- yolk sacs regardless of diet. This leaves the NAD Salvage Pathway, and its circulatory precursor NAM, as the main source of NAD in *Haao*^-/- yolk sacs, as well as their corresponding embryos, as suggested previously (*Cuny et al., 2023*; *Szot et al., 2024*). Indeed, NAM levels were lower in *Haao*^-/- yolk sacs compared to *Haao*^+/- under the equivalent diet, and yolk sac NAM levels strongly correlated with embryonic NAM in corresponding embryos irrespective of condition. Although not significantly different, the increase in NAM levels in both *Haao*^-/- tissues on the NW15 diet over the NW3.5 diet, and the corresponding significant increase in NAD levels themselves, likely originates from the additional provision of NAM from the mother under this diet. Correspondingly, levels of excretion products 2PY and 4PY, which are generated from NAM (*Figure 1A*) when its levels are in excess, were significantly lowered and close to zero in E10.5 *Haao*^-/- embryos on NW3.5 relative to NW15 embryos, indicating a low to insufficient NAD status.

Besides the direct effect of embryonic NAD deficiency as the cause of CNDD, it is possible that the low NAD levels observed in the yolk sac under NW3.5 diet also interfere with other crucial yolk sac functions that are linked to NAD, such as ATP-dependent active transport of essential nutrients (*Chang, 2010*; *Cindrova-Davies et al., 2017*; *Dutta and Sinha, 2017*) or NADP-dependent enzymes (*Di Pietro et al., 2002*; *Xiao et al., 2018*), thereby further impacting normal embryonic development. Likely, it is a combination of both. Determining the specific mechanisms of CNDD causation requires further studies.

Together, our NAD metabolome findings confirm that if vitamin B3 supply is limited, yolk sac NAD de novo synthesis becomes essential for maintaining NAD and normal embryo development. Further work with more refined analytical approaches, such as molecular flux analysis, could determine to which extent specific metabolites are exchanged between maternal circulation, the yolk sac, and the embryo.

## Yolk sac functions, very likely including NAD de novo synthesis, are conserved in human

Human embryogenesis requires a transient primary (day 7 to day 12) and secondary yolk sac (day 13 to day 49), while the mouse has a single yolk sac (E7.5 to birth). While human yolk sac anatomy and the timing of its presence differs to mouse, many functions are conserved between the species. These functions include serum protein synthesis, micro- and macronutrient transportation, and certain metabolic pathways, and it is also the site of primary haematopoiesis (*Cindrova-Davies et al., 2017*). Using published datasets, we showed that the human yolk sac expresses all genes required for NAD de novo synthesis during comparable embryonic stages as seen in the mouse. Therefore, it is plausible that yolk sac NAD de novo synthesis activity in early pregnancy is of similar importance to maintain NAD status and allow normal embryonic development in humans.

Recent studies have established a link between human yolk sac disruption and adverse pregnancy outcomes. Specifically, a disruption of yolk sac size and shape in the first trimester was a predictor of foetal anomalies and pregnancy loss (*Cho et al., 2006*; *Detti et al., 2021*; *Marin et al., 2021*; *Suguna and Sukanya, 2019*). Maternal diabetes during pregnancy is also a known risk factor for congenital anomalies (*Gabbay-Benziv et al., 2015*). Recent studies suggest that maternal diabetes not only affects the embryo but also the yolk sac due to hyperglycaemia, impairing nutrient transportation that drives embryopathy (*Dong et al., 2016*; *Reece et al., 1994*). Furthermore, diabetes has been

implicated in perturbations of the NAD de novo Synthesis Pathway (*Allegri et al., 2003*; *Liu et al., 2019*; *Oxenkrug, 2015*). Therefore, it is possible that disruption of NAD de novo synthesis, either maternally or in the yolk sac itself, may contribute to the formation of congenital malformations under diabetic conditions.

Furthermore, it is possible that the yolk sac NAD de novo synthesis capability is also conserved in non-mammalian vertebrates. At least one step of the pathway, catalysed by AFMID (*Figure 1A*), was found to be active in chicken egg yolk sac membranes (*Seifert, 2017*). Inhibition of this enzyme in the yolk sac resulted in lowered embryo NAD levels and some evidence of disrupted embryogenesis (*Seifert and Casida, 1978*; *Seifert and Casida, 1979*). Whether the entire NAD de novo Synthesis Pathway is active in the avian yolk sac, or whether downstream enzymatic synthesis steps are performed elsewhere, is unclear and requires further research.

## Outlook

We have shown in mice that NAD deficiency due to an impaired NAD de novo Synthesis Pathway can be prevented by supplementing pregnant mice with the NAD precursor vitamin B3 (*Cuny et al., 2020*; *Shi et al., 2017*; *Szot et al., 2024*). NAD deficiency between E7.5 and 10.5 recapitulates the majority of CNDD malformations, and under more severe conditions from E0.5 to 3.5 all embryos die (*Shi et al., 2017*). These mouse stages span approximately days 1–17 of human development (*Figure 6D*). Therefore, in women at risk of becoming NAD deficient, it would be necessary to enter pregnancy with sufficient NAD by taking vitamin B3 prior to pregnancy. Further research on the molecular intricacies of CNDD causation, the timing, and the tissues and cells involved will aid understanding of the causes of birth defects and miscarriages more generally.

Here, we highlight the importance of the yolk sac in embryogenesis. Given the link between yolk sac changes and miscarriage are established, other environmental and/or genetic factors may perturb yolk sac function and drive congenital malformation and embryo loss. Whilst CNDD should be preventable through supplementation, other causes may not be and instead require other approaches. Further research is required to better understand the mechanisms of CNDD causation and of other causes of adverse pregnancy outcomes involving the yolk sac.

# Materials and methods

### Key resources table

| Reagent type (species) or resource | Designation | Source or reference | Identifiers | Additional information |
|---|---|---|---|---|
| Strain, strain background (*Mus musculus*) | *Haao*em1Dunw | *Shi et al., 2017* | MGI:6285800 | |
| Sequence-based reagent | *Tdo2*_forward | This paper | PCR primers | AGGTGCTGCTCTGCTTGTTT |
| Sequence-based reagent | *Tdo2*_reverse | This paper | PCR primers | TGAGCGTGTCAATGTCCATAA |
| Sequence-based reagent | *Ido2*_forward | This paper | PCR primers | TGCCCTCAGACTTCCTCACT |
| Sequence-based reagent | *Ido2*_reverse | This paper | PCR primers | CGCTGCTCACGGTAACTCTT |
| Sequence-based reagent | *Afmid*_forward | This paper | PCR primers | AGCCACCTCCCAGAATGAC |
| Sequence-based reagent | *Afmid*_reverse | This paper | PCR primers | TGGAACCACATCCAAGTGTC |
| Sequence-based reagent | *Kmo*_forward | This paper | PCR primers | GAGCATTAACTTGGCCCTTTC |
| Sequence-based reagent | *Kmo*_reverse | This paper | PCR primers | AGTGGATCATTCTGGCTTTCA |

*Continued on next page*

*Continued*

| Reagent type (species) or resource | Designation | Source or reference | Identifiers | Additional information |
|---|---|---|---|---|
| Sequence-based reagent | *Kynu*_forward | This paper | PCR primers | GTTCAGTGGGCTGCACTTTT |
| Sequence-based reagent | *Kynu*_reverse | This paper | PCR primers | CCCAGTCATGTAAGCGGAGT |
| Sequence-based reagent | *Haao*_forward | This paper | PCR primers | ACAATGGGAGGGCAGTGTAT |
| Sequence-based reagent | *Haao*_reverse | This paper | PCR primers | CTTCTTACGGGCAGGGTCTT |
| Sequence-based reagent | *Qprt*_forward | This paper | PCR primers | ACTGGTGGAGAAGTATGGGC |
| Sequence-based reagent | *Qprt*_reverse | This paper | PCR primers | GGCTGCTACATTCCACCTCT |
| Sequence-based reagent | *Nadsyn1*_forward | This paper | PCR primers | GCCAAAGGCAAAGGTGCAAG |
| Sequence-based reagent | *Nadsyn1*_reverse | This paper | PCR primers | TAGCGAACATTCCGGTGCAT |
| Sequence-based reagent | *Nmnat1*_forward | This paper | PCR primers | GTGCCCAACTTGTGGAAGAT |
| Sequence-based reagent | *Nmnat1*_reverse | This paper | PCR primers | CAGCACATCGGACTCGTAGA |
| Sequence-based reagent | *Nmnat3*_forward | This paper | PCR primers | CCAGAGACCACCTACACCAAA |
| Sequence-based reagent | *Nmnat3*_reverse | This paper | PCR primers | CCACCCGAATCCAGTCAG |
| Sequence-based reagent | *Ubc*_forward | *Gu et al., 2014* | PCR primers | CCCAGTGTTACCACCAAG |
| Sequence-based reagent | *Ubc*_reverse | *Gu et al., 2014* | PCR primers | ATCACACCCAAGAACAAGC |
| Sequence-based reagent | *Ywhaz*_forward | *Jeong et al., 2014* | PCR primers | CAGTAGATGGAGAAAGATTTGC |
| Sequence-based reagent | *Ywhaz*_reverse | *Jeong et al., 2014* | PCR primers | GGGACAATTAGGGAAGTAAGT |
| Other | Rat and Mouse Premium Breeder Diet | Gordons Specialty Feeds, Yanderra, NSW, Australia | Cat. #: 27 | Mouse feed |
| Other | Tryptophan and niacin-free diet | Specialty Feeds, Glen Forrest, WA, Australia | Cat. #: SF21-083 | Mouse feed |
| Commercial assay or kit | Pierce BCA Protein Assay Kit | Thermo Fisher Scientific, Rockford, USA | Cat. #: 23225 | |
| Commercial assay or kit | PureLink RNA Mini kit | Thermo Fisher Scientific, Rockford, USA | Cat. #: 12183018 A | |
| Commercial assay or kit | Maxima First Strand cDNA Synthesis Kit | Thermo Fisher Scientific, Rockford, USA | Cat. #: K1642 | |
| Commercial assay or kit | LightCycler 480 SYBR Green I Master mix | Hoffmann-La Roche Ltd, Basel, Switzerland | Cat. #: 04887352001 | |
| Other | cOmplete Mini, EDTA-free Protease Inhibitor Cocktail | Merck KGaA, Darmstadt, Germany | Cat. #: 11836170001 | Protease inhibitors |
| Software, algorithm | Trimmomatic (v0.35) | *Bolger et al., 2014* | RRID:SCR_011848 | |

*Continued on next page*

*Continued*

| Reagent type (species) or resource | Designation | Source or reference | Identifiers | Additional information |
|---|---|---|---|---|
| Software, algorithm | STAR aligner (two pass method) | *Dobin et al., 2013* | RRID:SCR_004463 | |
| Software, algorithm | Seurat R package | *Butler et al., 2018* | https://satijalab.org/seurat/; RRID:SCR_016341 | |
| Software, algorithm | Prism (version 10) | GraphPad Software | RRID:SCR_002798 | |
| Software, algorithm | MetaboAnalyst 5.0 | *Pang et al., 2022* | RRID:SCR_015539 | |
| Software, algorithm | ShinyCell R package (v2.1) | *Ouyang et al., 2021* | https://github.com/ SGDDNB/ShinyCell; RRID:SCR_022756 | |
| Software, algorithm | R (v4.1) | R Project for Statistical Computing | https://www.r-project.org/; RRID:SCR_001905 | |

## Animal experiments

All animal experiments were performed in accordance with protocols approved by the Garvan Institute of Medical Research/St. Vincent's Animal Experimentation Ethics Committee, Sydney, Australia (approvals 18/27 and 21/18). All animal experiments were performed in accordance with relevant guidelines and regulations specified in the animal ethics approval. The *Haao* loss-of-function mouse line (allele *Haao*[em1Dunw]) has been described previously (*Shi et al., 2017*).

Mice were maintained on Breeder diet (Rat and Mouse Premium Breeder Diet, Gordons Specialty Feeds, Yanderra, NSW, Australia) containing 90 mg/kg niacin and 3.7 g/kg L-tryptophan. Mice were set up for timed matings and pregnancy was confirmed by the presence of a vaginal copulation plug in the morning (defined as timepoint E0.5). Pregnant females were culled between E10.5 and E18.5 by cervical dislocation and embryos dissected. From each embryo, the following tissues were collected: whole visceral yolk sac, placenta (whole placenta at E10.5–11.5, ½ placenta at E12.5–13.5, ¼ placenta at E14.5–18.5), and whole embryonic liver. All embryonic tissues were weighed prior to being snap frozen in liquid nitrogen. Tissues were genotyped for the *Haao* loss-of-functional allele as described previously (*Cuny et al., 2020*; *Shi et al., 2017*).

For experiments requiring limitation of the maternal dietary vitamin B3 supply during critical organogenesis stages, pregnant females were maintained on Breeder diet until E7.5. Between E7.5 and E10.5, the females were temporarily housed in a new cage and provided a diet lacking any NAD precursors (SF21-083, Specialty Feeds, Glen Forrest, WA, Australia) and drinking water containing 1 g/L L-tryptophan and 3.5 mg/L nicotinic acid (NW3.5) or 15 mg/L (NW15). At E10.5, females were re-placed on Breeder diet until embryo dissection.

E18.5 embryos were phenotyped as described previously (*Cuny et al., 2020*) except for the identification of heart defects, for which micro-CT was used. Briefly, hearts were dissected from fixed (4% paraformaldehyde with 1% glutaraldehyde) E18.5 embryos and stained in 0.7% phosphotungstic acid +30% ethanol in phosphate-buffered saline (PBS) for a minimum of 7 days or until stained completely. Stained hearts were scanned using a Skyscan 1272 scanner (Bruker Corporation, Billerica, USA) at a resolution of 1092x1632 with a voltage of 80 kV and a 125 µA current. The filter was set to Al 1 mm with a pixel size of 4.95 µm. Projection images were recorded with 2300 ms exposure time per projection with a rotation step of 0.4° between projections. The hearts were scanned at a total of 360° rotation. Acquired projections were reconstructed using Nrecon software (Bruker) with smoothing set to 3, ring artefact reduction to 19, and beam hardening to 10%. Reconstructed volumes were exported in TIFF format and visually inspected for structural heart defects.

## HAAO enzyme assays

HAAO enzyme assays were performed as previously described, with some minor modifications (*Shi et al., 2017*; *Walsh et al., 1991*). Briefly, volume of tissue lysis buffer (10 mM protease inhibitor in PBS) was adjusted by stage and tissue type to standardise total protein levels per assay (see *Supplementary file 1*). Due to the small tissue size, yolk sacs and embryonic livers were pooled at early gestational stages to obtain sufficient material per biological replicate. Regardless of tissue type, 15 µL of

tissue lysate was used per reaction and normalised by total protein per reaction. Protein concentration was determined by Pierce BCA Protein Assay Kit (Thermo Scientific, Rockford, USA), performed according to the manufacturer's protocol.

## Quantitative real time PCR (qRT-PCR)

RNA was extracted from snap frozen *Haao*^+/- yolk sacs using PureLink RNA Mini kit (Thermo Scientific, Rockford, USA), as per the manufacturer's instructions. Tissue homogenization was performed in 650 μL lysis buffer using a Bio-Gen Pro 200 homogenizer (PRO Scientific Inc, Oxford, Connecticut, USA). cDNA was synthesised from 1 μg RNA using Maxima First Strand cDNA Synthesis Kit (K1672, Thermo Scientific) according to the manufacturer's protocol, with the addition of a dilution step to 1.6 ng/μL in water. qPCR was performed on CFX384 Touch Real-Time PCR Detection System (Bio-Rad, USA) using the primers detailed in *Supplementary file 7* and LightCycler 480 SYBR Green I Master mix (04887352001; F. Hoffmann-La Roche Ltd, Basel, Switzerland). Relative expression was calculated using the ΔCt method, normalizing each gene to both *Ubc* and *Ywhaz* levels.

## Metabolite quantification by UHPLC-MS/MS

NAD^+ and related metabolites were quantified in yolk sacs and placentas collected at E10.5, E12.5 and E14.5, in embryos collected at E10.5, and in embryonic livers collected at E14.5 using UHPLC-MS/MS as described for mouse liver tissue (*Cuny et al., 2021*), with a minor adjustment to improve quantification of high concentration metabolites. From the last supernatant after addition of chloroform and centrifugation, 30 μL were taken, diluted with 70 μL of 100 mM ammonium acetate in water (=sample for abundant metabolite quantification), and measured. The remainder of supernatant was dried in a vacuum centrifuge at room temperature, metabolites reconstituted in 60 μL of 100 mM ammonium acetate, centrifuged for 2 min at 15,000 × *g* and 4 °C, and 50 μL of the supernatant taken and measured by UHPLC-MS/MS as described (=sample for low concentration metabolite quantification). Metabolite concentration values were normalised by the tissue sample wet weight at dissection. Metabolite concentrations in NW3.5 and NW15 embryos and yolk sacs were additionally normalised by protein concentration in the solid fraction (pellet) remaining after the first centrifugation step of the metabolite extraction procedure. To determine total protein concentration in the pellets, they were washed in methanol, air dried using a vacuum centrifuge, resuspended in 0.2 M sodium hydroxide, and incubated at 95 °C for 20 min. Insoluble proteins and other debris were removed by centrifugation at 12,000 rpm for 7 min in a microcentrifuge and protein concentration was determined using the Pierce BCA Protein Assay Kit (Thermo Fisher Scientific, Rockford, USA) according to the manufacturer's protocol.

## Analysis of published RNA-seq datasets

RNA-Seq datasets (matrix tables for single cell and fastq for bulk RNA-Seq) were downloaded from public repositories (*Supplementary file 8*). For the re-analysis of bulk RNA sequencing fastq files were trimmed of low quality sequence and adaptor using Trimmomatic (v0.35) (*Bolger et al., 2014*) before being aligned to hg38 genome using the STAR aligner (two pass method) (*Dobin et al., 2013*). Gene counts were generated from the STAR aligner using gencode vM10 as the reference gene model. Gene count matrixes were filtered to only contain genes expressed at 1 count per million (CPM) in at least 3 samples before being normalised by Trimmed Mean of M-values (TMM) and log2 transformed using R scripts.

For the re-analysis of single cell datasets, single cell count matrixes were downloaded and assembled into Seurat objects using the Seurat R package (*Butler et al., 2018*). Cells were filtered for percent of mitochondrial content, and clustering included detection of top variable genes, scaling, dimensionality reduction with PCA and construction of nearest neighbour graphs. Cell cluster identification was established from markers described in linked manuscript (*Supplementary file 8*). For the gastrulation atlas, single cell count objects were downloaded from the Bioconductor 'MouseGastrulationData' resource (*Griffiths and Lun, 2023*). Seurat objects were converted into ShinyCell web applets (*Ouyang et al., 2021*) to help visualise and explore the data.

## Statistical analysis

One-way ANOVA was used to determine statistical significance of tissue sample gene expression levels at different developmental stages, of HAAO activity in tissues of different *Haao* genotypes,

and of metabolite concentrations under different *Haao* genotypes and maternal diets. Significance of metabolite level differences was additionally analysed by one-way ANOVA with Tukey's multiple comparisons test. Student's t-test was used to determine statistical difference of *Haao*$^{+/-}$ and *Haao*$^{-/-}$ embryo weights with the NW3.5 diet. Values are displayed as mean ± standard deviation. All statistical analyses were performed with Prism (version 10; GraphPad Software). Metaboanalyst 5.0 (*Pang et al., 2022*) was used to generate PCA plots and heatmaps of the NAD metabolome data. Custom R scripts were used to generate heatmaps of gene expression data.

## Acknowledgements

This research is supported by funds to SLD from: the National Health and Medical Research Council (NHMRC), Principal Research Fellowship (ID1135886), Leadership Level 3 Fellowship (ID2007896) and Project Grant (ID1162878); a NSW Health Cardiovascular Research Capacity Program Senior Researcher Grant; and philanthropic support from the Key Foundation. We gratefully acknowledge the Victor Chang Cardiac Research Institute Innovation Centre, funded by the NSW Government, as well as funding from the Freedman Foundation for the Metabolomics Facility. We thank Joelene Greasby for technical assistance.

## Additional information

### Funding

| Funder | Grant reference number | Author |
|---|---|---|
| National Health and Medical Research Council | ID1135886 | Sally L Dunwoodie |
| National Health and Medical Research Council | ID2007896 | Sally L Dunwoodie |
| National Health and Medical Research Council | ID1162878 | Sally L Dunwoodie |
| NSW Health | Cardiovascular Research Capacity Program Senior Researcher Grant | Sally L Dunwoodie |
| Key Foundation | | Sally L Dunwoodie |

The funders had no role in study design, data collection and interpretation, or the decision to submit the work for publication.

### Author contributions

Kayleigh Bozon, Conceptualization, Data curation, Formal analysis, Validation, Investigation, Visualization, Methodology, Writing – original draft, Writing – review and editing; Hartmut Cuny, Conceptualization, Data curation, Formal analysis, Supervision, Validation, Investigation, Visualization, Writing – original draft, Writing – review and editing; Delicia Z Sheng, Ella MMA Martin, Alena Sipka, Data curation, Formal analysis, Writing – review and editing; Paul Young, David T Humphreys, Validation, Investigation, Methodology, Writing – review and editing; Sally L Dunwoodie, Conceptualization, Resources, Supervision, Funding acquisition, Validation, Writing – original draft, Project administration, Writing – review and editing

### Author ORCIDs

Hartmut Cuny (ID) https://orcid.org/0000-0002-1551-2354
David T Humphreys (ID) https://orcid.org/0000-0003-4140-0089
Sally L Dunwoodie (ID) https://orcid.org/0000-0002-2069-7349

### Ethics

All animal experiments were performed in accordance with protocols approved by the Garvan Institute of Medical Research/St. Vincent's Animal Experimentation Ethics Committee, Sydney, Australia (approvals 18/27 and 21/18). All animal experiments were performed in accordance with relevant guidelines and regulations specified in the animal ethics approval.

Reviewer #1 (Public review): https://doi.org/10.7554/eLife.97649.3.sa1
Reviewer #2 (Public review): https://doi.org/10.7554/eLife.97649.3.sa2
Author response https://doi.org/10.7554/eLife.97649.3.sa3

## Additional files

### Supplementary files

Supplementary file 1. HAAO enzyme activity in embryonic liver, yolk sac, placenta, and embryo at different stages in gestation.

Supplementary file 2. NAD metabolite concentrations in embryonic liver at E14.5 by *Haao* genotype, as measured by UHPLC-MS/MS.

Supplementary file 3. NAD metabolite concentrations in the visceral yolk sac at E10.5, E12.5 and E14.5 by *Haao* genotype, as measured by UHPLC-MS/MS.

Supplementary file 4. Summary of congenital malformation types and incidence in embryos at E18.5 arising from maternal NW3.5 diet provision between E7.5 and E10.5.

Supplementary file 5. NAD metabolite concentrations in the E10.5 embryo and yolk sac with maternal NW15 or NW3.5 diet provision, normalised by protein.

Supplementary file 6. NAD metabolite concentrations in the E10.5 embryo and yolk sac with maternal NW15 or NW3.5 diet provision, normalised by wet weight.

Supplementary file 7. Primers for RT-qPCR.

Supplementary file 8. Summary of published RNA-seq datasets used and assessed in this study.

MDAR checklist

### Data availability

All data generated or analysed during this study are included in the manuscript.

The following previously published datasets were used:

| Author(s) | Year | Dataset title | Dataset URL | Database and Identifier |
|---|---|---|---|---|
| Cindrova-Davies T, Jauniaux E, Elliot MG, Gong S, Burton GJ, Charnock-Jones DS | 2017 | RNASeq reveals conservation of function among the yolk sacs of human, mouse and chicken | https://www.ebi.ac.uk/ena/browser/view/PRJEB18767 | European Nucleotide Archive, PRJEB18767 |
| Zhao H, Choi K | 2019 | Single cell RNA-seq of mouse embryoid bodies and yolk sac | https://www.ncbi.nlm.nih.gov/geo/query/acc.cgi?acc=GSM3732840 | NCBI Gene Expression Omnibus, GSM3732840 |
| Zhong Y | 2018 | Landscape of Endoderm and Liver Developmental Lineage Revealed by Single-Cell RNA Sequencing | https://db.cngb.org/search/project/CNP0000236/ | China National GeneBank DataBase, CNP0000236 |
| Marsh B | 2020 | Single nuclei RNA-seq of mouse placental labyrinth development | https://doi.org/10.6084/m9.figshare.13204265.v1 | figshare, 10.6084/m9.figshare.13204265.v1 |
| Griffiths J, Lun A | 2022 | MouseGastrulationData datasets | https://bioconductor.org/packages/devel/data/experiment/vignettes/MouseGastrulationData/inst/doc/MouseGastrulationData.html | Bioconductor, MouseGastrulationDatapackage |

*Continued on next page*

*Continued*

| Author(s) | Year | Dataset title | Dataset URL | Database and Identifier |
|---|---|---|---|---|
| Goh i, et al | 2023 | EL_YS_main_combined_log_norm | https://app.cellatlas.io/yolk-sac/dataset/34/scatterplot | CellAtlas.io, EL_YS_main_combined_log_norm_20220706.zarr |
| Wesley, et al | 2022 | Hepatocytes_primary | https://app.cellatlas.io/liver-development/dataset/18/scatterplot | CellAtlas.io, Hepatocytes_primary_ad.zarr |
| Cao, et al | 2019 | Raw Gene Count Matrices and Metadata | https://oncoscape.v3.sttrcancer.org/atlas.gs.washington.edu.mouse.rna/downloads | Mouse RNA Atlas, MouseAtlas |

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
