## [Editor Report · eLife Assessment]

NAD deficiency perturbs embryonic development resulting in multiple congenital malformations, collectively termed Congenital NAD Deficiency Disorder (CNDD). The authors report **fundamental** findings demonstrating that extra-embryonic visceral yolk sac endoderm is critical for NAD de novo synthesis during early organogenesis and perturbations of this pathway may underlie CNDD. The authors combine gene expression with metabolic assays to provide **solid** evidence of an essential role of the extra-embryonic visceral yolk sac in both mouse and human embryos.

---

## [Referee Report · Reviewer #1 (Public review)]

Summary:

This study investigated the mechanism underlying Congenital NAD Deficiency Disorder (CNDD) using a mouse model with loss of function of the HAAO enzyme which mediates a key step in the NAD de novo synthesis pathway. This study builds on the observation that the kynurenine pathway is required in the conceptus, as HAAO null embryos are sensitive to maternal deficiency of NAD precursors (vitamin B3) and tryptophan, and narrows the window of sensitivity to a 3 day period.

An important finding is that de novo NAD synthesis occurs in an extra-embryonic tissue, the visceral yolk sac, before the liver develops in the embryo. It is suggested that lack of this yolk sac activity leads to impaired NAD supply in the embryo leading to structural abnormalities found later in development.

Strengths:

Previous studies show a requirement for HAAO activity for normal development of the embryos develop abnormalities under conditions of maternal vitamin B3 deficiency, indicating a requirement for NAD synthesis in the conceptus. Analysis of scRNA-seq datasets combined with metabolite analysis of yolk sac tissue shows that the NAD synthesis pathway is expressed and functional in the yolk sac from E10.5 onwards (prior to liver development).

HAAO enzyme assay enabled quantification of enzyme activity in relevant tissues including liver (from E12.5), embryo, placenta and yolk sac (from E11.5).

Comprehensive metabolite analysis of the NAD synthesis pathway supports the predicted effects of HAAO knockout and provides analysis of yolk sac, placenta and embryo at a series of stages.

The dietary study (with lower vitamin B3 in maternal diet from E7.5-10.5) is an incremental addition to previous studies which imposed similar restrictions from E7.5-12.5. Nevertheless, this emphasises the importance of the synthesis pathway on the conceptus at stages before liver activity is prominent.

Weaknesses:

The current dietary study narrows the period when deficiency can cause malformations (analysed at E18.5), and altered metabolite profiles (eg, increased 3HAA, lower NAD) are detected in yolk sac and embryo at E10.5.

More importantly, there is still a question of whether in addition to the yolks sac, there is HAAO activity within the embryo itself has been assayed as early as E11.5, with minimal activity prior to E12.5 (when it is assayed in liver). These findings support the hypothesis that within the conceptus (embryo, chorioallantoic placenta and visceral yok sac) the embryo is unlikely to be the site of NAD synthesis prior to liver development.

Evidence for lack of function of the NAD synthesis pathway in the embryos itself from kynurenine at E7.5-10.5 comes from reanalysis of scRNA-seq. This suggests low or absent expression of HAAO in the embryo prior to E10.5 (corresponding to the period when the authors have demonstrated that de novo NAD synthesis in the conceptus is needed). The caveat to this conclusion is that additional analysis of RNA and/or protein expression in the embryos at E7.5-10.5 has not been performed to validate the scRNA-seq data.

---

## [Referee Report · Reviewer #2 (Public review)]

Summary:

Disruption of the nicotinamide adenine dinucleotide (NAD) de novo Synthesis Pathway, by which L-tryptophan is converted to NAD results in multi-organ malformations which collectively has been termed Congenital NAD Deficiency Disorder (CNDD).

While NAD de novo synthesis is primarily active in the liver postnatally, the site of activity prior to and during organogenesis is unknown. However, mouse embryos are susceptible to CNDD between E7.5-E12.5, before the embryo has developed a functional liver. Therefore, NAD de novo synthesis is likely active in another cell or tissue during this time window of susceptibility.

The body of work presented in this paper continues the corresponding author's labs investigation of the cause and effects of NAD Deficiency and the primary goal was to determine the cell or tissue responsible for NAD de novo synthesis during early embryogenesis.

The authors conclude that visceral yolk sac endoderm is the source of NAD de novo synthesis, which is essential for mouse embryonic development, and furthermore that the dynamics of NAD synthesis are conserved in human equivalent cells and tissues, the perturbation of which results in CNDD.

Strengths:

Overall, the primary findings regarding the source of NAD synthesis, the temporal requirement and conservation between rodent and human species is quite novel and important for our understanding of NAD synthesis and function and role in CNDD.

The authors used UHPLC-MS/MS to quantify NAD+ and NAD-related metabolites and showed convincingly that the NAD salvage pathway can compensate for the loss of NAD synthesis in Haao-/- embryos, then determined that Haao activity was present in the yolk sac prior to hepatic development identifying this organ as the site of de novo NAD synthesis. Dietary modulation between E7.5-10.5 was sufficient to induce CNDD phenotypes, narrowing the window of susceptibility, and then re-analysis of RNA-seq datasets suggested the endoderm was the cell source of NAD synthesis.

Weaknesses:

Page 4 and Table S4. The descriptors for malformations of organs such as the kidney and vertebrae are quite vague and uninformative. More specific details are required to convey the type and range of anomalies observed as a consequence of NAD deficiency.

Can the authors define whether the role for the NAD pathway in a couple of tissue or organ systems is the same. By this I mean is the molecular or cellular effect of NAD deficiency the same in the vertebrae and organs such as the kidney. What unifies the effects on these specific tissues and organs and are all tissues and organs affected. If some are not, can the authors explain why they escape the need for the NAD pathway.

Page 5 and Figure 6C. The expectation and conclusion for whether specific genes are expressed in particular cell types in scRNA-seq datasets depends on number of cells sequenced, the technology (methodology) used, the depth of sequencing and also the resolution of the analysis. It is therefore essential to perform secondary validation of the analysis of scRNA-seq data. At a minimum, the authors should perform in situ hybridization or immunostaining for Tdo2, Afmid, Kmo, Kynu, Haao, Qprt and Nadsyn1 or some combination thereof at multiple time points during early mouse embryogenesis to truly understand the spatiotemporal dynamics of expression and NAD synthesis.

Absolute functional proof of the yolk sac endoderm as being essential and required for NAD synthesis in the context of CNDD might require conditional deletion of Haao in the yolk sac versus embryo using appropriate Cre driver lines or in the absence of a conditional allele, could be performed by tetraploid embryo-ES cell complementation approaches. But temporal dietary intervention can also approximate the same thing by perturbing NAD synthesis then the yolk sac is the primary source versus when the liver becomes the primary source in the embryo.

In further revisions, the authors have added data to Supp Table 4 and Supplemental Figures 1 and 2

Although the authors did not perform in situ hybridization for some of the genes requested to define the critical cell type of expression, available scRNA-sequencing suggests the yolk sac endoderm are the only likely source of NAD synthesis prior to its synthesis in the liver. Absolute functional proof of the yolk sac endoderm as being essential and required for NAD synthesis in the context of CNDD still requires validation but nonetheless it seems likely given the absence of a functional liver in embryos prior to E12.5. The authors provided some additional data pertaining to the type of kidney and vertebral anomalies observed which makes this data more complete.

---

## [Author Response]

The following is the authors’ response to the current reviews.

**Recommendations for the authors:**

**Reviewer #1 (Recommendations for the authors):**
A number of modifications/additions have been made to the text which help to clarify the background and details of the study and I feel have improved the study.NAD deficiency induced using the dietary/Haao null model showed a window of susceptibility at E7.5-10.5. Further, HAAO enymze activity data has been added at E11.5 and the minimal HAAO activity in the embryo act E11.5 supports the hypothesis that the NAD synthesis pathway from kynurenine is not functional until the liver starts to develop.The caveat to this is that absence of expression/activity in embryonic cells at E7.5-10/5 relies on previous scRNA-seq data. Both reviewers commented that analysis of RNA and/or protein expression at these stages (E7.5-10.5) would be necessary to rule this out, and would strongly support the conclusions regarding the necessity for yolk sac activity.There are a number of antibodies for HAAO, KNYU etc so it is surprising if none of these are specific for the mouse proteins, while an alternative approach in situ hydridisation would also be possible.

We have tested 2 anti-HAAO antibodies, 2 anti-KYNU antibodies and 1 anti-QPRT antibody on adult liver and various embryonic tissues. Given that all tested antibodies only detected a specific band in tissues with very high expression and abundant target protein levels (adult liver), they were determined to be unsuitable to conclusively prove that these proteins of the NAD de novo synthesis pathway are absent in embryos prior to the development of a functional liver. They were also unsuitable for IHC experiments to determine which cell types (if any) have these proteins.

The antibodies, tested assays and samples, and the results obtained were as follows:

**Anti-HAAO antibody (ab106436, Abcam, UK):**

Was tested in western blots of liver, E11.5-E14.5 yolk sac, E14.5 placenta, and E14.5 and E16.5 embryonic liver lysates from wild-type (WT) and *Haao*-/- mice. The target band (32.5 KD) was visible in the WT liver samples and absent in *Haao*-/- livers, and faintly visible in E11.5-E14.5 WT yolk sac, with intensity gradually increasing in E12.5 and E13.5 WT yolk sac. Multiple strong non-specific bands occurred in all samples, requiring cutting off the >50 KD area of the blots.

Was re-tested in western blots comparing WT, *Haao*-/-, and *Kynu*-/- E9.5-E11.5 embryo, E9.5 yolk sac, and adult liver tissues. It detected the target band faintly only in WT and *Kynu*-/- liver lysates. No target band could be resolved in E9.5 yolk sac or embryo lysates. Due to the low sensitivity of the antibody, it is unsuitable to conclusively determine whether HAAO is present or absent in E9.5 yolk sacs and E9.5-E11.5 embryos.

Was tested in IHC with DAB and IF, producing non-specific staining on both WT and *Haao*-/- liver and kidney tissue.

**Anti-HAAO antibody (NBP1-77361, Novus Biologicals, LLC, CO, USA):**

Was tested in western blots and detected a very faint target band in WT liver lysate that was absent in *Haao*-/- lysate, with stronger non-specific bands occurring in both genotypes.

Was tested in IHC with DAB, producing non-specific staining on both WT and *Haao*-/- liver and kidney tissue.

**Anti-L-Kynurenine Hydrolase antibody (11796-1-AP, Proteintech Group, IL, USA):**

Was tested in western blots and detected a faint target band (52 KD) in E11.5, E12.5 E13.5, and E14.5 yolk sac lysates. Detected a weak band in E14.5 liver, a stronger band in E16.5 liver, but not in E14.5 placenta. The target band was only resolved with normal ECL substrate and extended exposure when the >75 KD part of the blot was cut off.

Was re-tested in western blots comparing WT, *Haao*-/-, and *Kynu*-/- E9.5-E11.5 embryo, E9.5 yolk sac, and adult liver tissues. It detected the target band only in WT and *Haao*-/- liver lysates, requiring Ultra Sensitive Substrate. No target band could be resolved in yolk sac or embryo lysates of any genotype.

**Anti-L-Kynurenine Hydrolase antibody (ab236980, Abcam, UK):**

Was tested in western blots and detected a very faint target band (52 KD) in WT liver lysates and no band in *Kynu*-/- liver lysates. Multiple non-specific bands occurred irrespective of the *Kynu* genotype of the lysate.

Was tested in IHC with DAB and IF, producing non-specific staining on both WT and *Kynu*-/- liver and kidney tissue.

**Anti-QPRT (orb317756, Biorbyt, NC, USA):**

Was tested in western blots and detected a faint target band (31 KD) with multiple other bands between 25-75 KD and an extremely strong band around 150 KD on WT liver lysates.

The following is the authors’ response to the original reviews.

**Reviewer 1 Public Review:**
The current dietary study narrows the period when deficiency can cause malformations (analysed at E18.5), and altered metabolite profiles (eg, increased 3HAA, lower NAD) are detected in the yolk sac and embryo at E10.5. However, without analysis of embryos at later stages in this experiment it is not known how long is needed for NAD synthesis to be recovered - and therefore until when the period of exposure to insufficient NAD lasts. This information would inform the understanding of the developmental origin of the observed defects.

Our previous published work (Cuny et al 2023 https://doi.org/10.1242/dmm.049647) indicates that the timing of NAD de novo synthesis pathway precursor availability and consequently the timing of NAD deficiency during organogenesis drives which organs are affected in their development. Furthermore, experimental data of another project (manuscript submitted) shows that mouse embryos (from mothers on an NAD precursor restricted diet that induces CNDD) were NAD deficient at E9.5 and E11.5, but embryo NAD levels were fully recovered at E14.5 when compared to same-stage embryos from mothers on precursor-sufficient diet. This was observed irrespective of the embryos’ *Haao* genotype. In the current study, NAD precursor provision was only restricted until E10.5. Thus, we expect that our embryos phenotyped at E18.5 had recovered their NAD levels back to normal by E14.5 at the latest. More research, beyond the scope of the current manuscript, is required to spatio-temporally link embryonic NAD deficiency to the occurrence of specific defect types and elucidate the mechanistic origin of the defects. To acknowledge this, we updated the respective Discussion paragraph on page 7 and added the following statement: “This observation supports our hypothesis that the timing of NAD deficiency during organogenesis determines which organs/tissues are affected (Cuny et al., 2023), but more research is needed to fully characterise the onset and duration of embryonic NAD deficiency in dietary NAD precursor restriction mouse models.”

More importantly, there is still a question of whether in addition to the yolk sac, there is HAAO activity within the embryo itself prior to E12.5 (when it has first been assayed in the liver - Figure 1C). The prediction is that within the conceptus (embryo, chorioallantoic placenta, and visceral yok sac) the embryo is unlikely to be the site of NAD synthesis prior to liver development. Reanalysis of scRNA-seq (Fig 1B) shows expression of all the enzymes of the kynurenine pathway from E9.5 onwards. However, the expression of another available dataset at E10.5 (Fig S3) suggested that expression is 'negligible'. While the expression in Figure 1B, Figure S1 is weak this creates a lack of clarity about the possible expression of HAAO in the hepatocyte lineage, or especially elsewhere in the embryo prior to E10.5 (corresponding to the period when the authors have demonstrated that de novo NAD synthesis in the conceptus is needed). Given these questions, a direct analysis of RNA and/or protein expression in the embryos at E7.5-10.5 would be helpful.

We now have included additional data showing that whole embryos at E11.5 and embryos with their livers removed at E14.5 have negligible HAAO enzyme activity. The observed lack of HAAO activity in the embryo at E11.5 is consistent with the absence of a functional embryonic liver at that stage. Thus, it confirms that the embryo is dependent of extraembryonic tissues (the yolk sac) for NAD de novo synthesis prior to E12.5. The additional datasets are now included in Supplementary Table S1 and as Supplementary Figure 2. The Results section on page 2 has been updated to refer to these datasets.

**Reviewer #2 (Public Review):**
Page 4 and Table S4. The descriptors for malformations of organs such as the kidney and vertebrae are quite vague and uninformative. More specific details are required to convey the type and range of anomalies observed as a consequence of NAD deficiency.

We now provide more information about the malformation types in the Results on page 4. Also, Table S4 now defines the missing vertebral, sternum, and kidney descriptors.

Can the authors define whether the role of the NAD pathway in a couple of tissue or organ systems is the same? By this I mean is the molecular or cellular effect of NAD deficiency is the same in the vertebrae and organs such as the kidney. What unifies the effects on these specific tissues and organs and are all tissues and organs affected? If some are not, can the authors explain why they escape the need for the NAD pathway?

This is a good comment, highlighting that further research, beyond the scope of this manuscript, is needed to better understand the underlying mechanisms of CNDD causation. We have expanded the Discussion paragraph “NAD deficiency in early organogenesis is sufficient to cause CNDD” to indicate that while the timing of NAD deficiency during embryogenesis explains variability in phenotypes among the CNDD spectrum, it is unknown why other organs/tissues are seemingly not affected by NAD deficiency.

To answer the reviewer’s questions and elucidate the underlying cellular and molecular processes in individual organs affected by NAD deficiency, a multiomic approach is required. This is because NAD is involved in hundreds of molecular and cellular processes affecting gene expression, protein levels, metabolism, etc. For details of NAD functions that have relevance to embryogenesis, the reviewer may refer to our recent review article (Dunwoodie et al 2023 https://doi.org/10.1089/ars.2023.0349).

Page 5 and Figure 6C. The expectation and conclusion for whether specific genes are expressed in particular cell types in scRNA-seq datasets depend on the number of cells sequenced, the technology (methodology) used, the depth of sequencing, and also the resolution of the analysis. It is therefore essential to perform secondary validation of the analysis of scRNA-seq data. At a minimum, the authors should perform in situ hybridization or immunostaining for Tdo2, Afmid, Kmo, Kynu, Haao, Qprt, and Nadsyn1 or some combination thereof at multiple time points during early mouse embryogenesis to truly understand the spatiotemporal dynamics of expression and NAD synthesis.

We have tested antibodies against HAAO, KYNU, and QPRT in adult mouse liver samples (the main site of NAD de novo synthesis) but these produced non-specific bands in western blotting experiments. Therefore, immunostaining studies on embryonic tissues were not feasible.

However, we agree that histological methods such as in situ hybridisation would provide secondary validation of the exact cell types that express these genes. To acknowledge this, we have updated a sentence on page 5 referring to the data shown in Figure 6C as follows: “While histological methods such as in situ hybridisation would be required to confirm the exact cell types expressing these genes, the available expression data indicates that the genes encoding those enzymes required to convert L-kynurenine to NAD (kynurenine pathway) are exclusively expressed in the yolk sac endoderm lineage from the onset of organogenesis (E8.0-8.5).”

Absolute functional proof of the yolk sac endoderm as being essential and required for NAD synthesis in the context of CNDD might require conditional deletion of Haao in the yolk sac versus embryo using appropriate Cre driver lines or in the absence of a conditional allele, could be performed by tetraploid embryo-ES cell complementation approaches. But temporal dietary intervention can also approximate the same thing by perturbing NAD synthesis Shen the yolk sac is the primary source versus when the liver becomes the primary source in the embryo.

Reviewer 1 has made a similar comment about confirming that indeed NAD de novo synthesis activity is limited to extraembryonic tissues (=yolk sacs) and absent in the embryo prior to development of an embryonic liver. We now have included additional data showing that whole embryos at E11.5 and embryos with their livers removed at E14.5 have negligible HAAO enzyme activity. The observed lack of HAAO activity in the embryo at E11.5 is consistent with the absence of a functional embryonic liver at that stage. We think this provides enough proof that the embryo is dependent of extraembryonic tissues (the yolk sac) for NAD de novo synthesis prior to E12.5. The additional datasets are now included in Supplementary Table S1 and as Supplementary Figure 2. The Results section on page 2 has been updated to refer to these data.

**Reviewer #1 (Recommendations For The Authors):**
(1) Introduction (page 1) introduces mouse models with defects in the kynurenine pathway "confirming that NAD de novo synthesis is required during embryogenesis ...". This requirement is revealed by the imposition of maternal dietary deficiency and more detail (or a more clear link to the following sentences) here would help the reader who is not familiar with the previous papers using the HAAO mice and dietary modulation.

We have updated this paragraph in the Introduction to better indicate that the requirement of NAD de novo synthesis for embryogenesis was confirmed in mouse models by modulating the maternal dietary NAD precursor provision during pregnancy.

(2) Discussion - throughout the introduction and results the authors refer to the NAD de novo synthesis pathway, with the study focussing on the effects of HAAO loss of function. Data implies that the kynurenine pathway is active in the yolk sac but whether de novo synthesis from L-tryptophan occurs has not been addressed. The first sub-heading of the discussion could be more accurate referring to the kynurenine pathway, or synthesis from kynurenine.

We agree that our manuscript needed to make better distinction between NAD de novo synthesis starting from kynurenine and starting from tryptophan. We removed “from Ltryptophan” from the sub-heading in the Discussion and clarified in this paragraph which genes are required to convert tryptophan to kynurenine and which genes to convert kynurenine to NAD. We also updated two Results paragraphs (page 2, 2nd paragraph; page 5, 5th paragraph) to improve clarity.

It is worth noting that our statement in the Discussion “this is the first demonstration of NAD de novo synthesis occurring in a tissue outside of the liver and kidney.” is valid because vascular smooth muscle cells express *Tdo2* and in combination with the other requisite genes expressed in endoderm cells, the yolk sac has the capability to synthesise NAD de novo from L-tryptophan.

(3) Outlook - While this section is designed to be looking ahead to the potential implications of the work, the last section on gene therapy of the yolk sac seems far removed from the paper content and highly speculative. I feel this could detract from the main points of the study and could be removed.

We have updated the Outlook paragraph and shortened the final part to “Further research is required to better understand the mechanisms of CNDD causation and of other causes of adverse pregnancy outcomes involving the yolk sac.”

(4) In Figure 2D it would be useful to label the clusters as the colours in the legend are difficult to match to the heatmap.

We now have labelled the clusters with lowercase letters above the heatmap to make it easier to match the clusters in Figure 2D to the colours used for designating tissues and genotypes. These labels are described in the figure’s key and the figure legend.

**Reviewer #2 (Recommendations For The Authors):**
Page 4 and Table S4. The descriptors for malformations of organs such as the kidney and vertebrae are quite vague and uninformative. More specific details are required to convey the type and range of anomalies observed as a consequence of NAD deficiency.

We now provide more information about the malformation types in the Results on page 4. Also, Table S4 now defines the missing vertebral, sternum, and kidney descriptors.

Can the authors define whether the role of the NAD pathway in a couple of tissue or organ systems is the same? By this I mean is the molecular or cellular effect of NAD deficiency is the same in the vertebrae and organs such as the kidney. What unifies the effects on these specific tissues and organs and are all tissues and organs affected? If some are not, can the authors explain why they escape the need for the NAD pathway?

This is a good comment, highlighting that further research, beyond the scope of this manuscript, is needed to better understand the underlying mechanisms of CNDD causation. We have expanded the Discussion paragraph “NAD deficiency in early organogenesis is sufficient to cause CNDD” to indicate that while the timing of NAD deficiency during embryogenesis explains variability in phenotypes among the CNDD spectrum, it is unknown why other organs/tissues are seemingly not affected by NAD deficiency.

To answer the reviewer’s questions and elucidate the underlying cellular and molecular processes in individual organs affected by NAD deficiency, a multiomic approach is required. This is because NAD is involved in hundreds of molecular and cellular processes affecting gene expression, protein levels, metabolism, etc. For details of NAD functions that have relevance to embryogenesis, the reviewer may refer to our recent review article (Dunwoodie et al 2023 https://doi.org/10.1089/ars.2023.0349).

Page 5 and Figure 6C. The expectation and conclusion for whether specific genes are expressed in particular cell types in scRNA-seq datasets depend on the number of cells sequenced, the technology (methodology) used, the depth of sequencing, and also the resolution of the analysis. It is therefore essential to perform secondary validation of the analysis of scRNA-seq data. At a minimum, the authors should perform in situ hybridization or immunostaining for Tdo2, Afmid, Kmo, Kynu, Haao, Qprt, and Nadsyn1 or some combination thereof at multiple time points during early mouse embryogenesis to truly understand the spatiotemporal dynamics of expression and NAD synthesis.

We have tested antibodies against HAAO, KYNU, and QPRT in adult mouse liver samples (the main site of NAD de novo synthesis) but these produced non-specific bands in western blotting experiments. Therefore, immunostaining studies on embryonic tissues were not feasible.

However, we agree that histological methods such as in situ hybridisation would provide secondary validation of the exact cell types that express these genes. To acknowledge this, we have updated a sentence on page 5 referring to the data shown in Figure 6C as follows: “While histological methods such as in situ hybridisation would be required to confirm the exact cell types expressing these genes, the available expression data indicates that the genes encoding those enzymes required to convert L-kynurenine to NAD (kynurenine pathway) are exclusively expressed in the yolk sac endoderm lineage from the onset of organogenesis (E8.0-8.5).”